# Hypoxia-induced mobilization of NHE6 to the plasma membrane triggers endosome hyperacidification and chemoresistance

Fabrice Lucien[1], Pierre-Paul Pelletier[1], Roxane R. Lavoie[1], Jean-Michel Lacroix[1], Sébastien Roy[2], Jean-Luc Parent[2], Dominique Arsenault[1], Kelly Harper[1] & Claire M. Dubois[1]

The pH-dependent partitioning of chemotherapeutic drugs is a fundamental yet understudied drug distribution mechanism that may underlie the low success rates of current approaches to counter multidrug resistance (MDR). This mechanism is influenced by the hypoxic tumour microenvironment and results in selective trapping of weakly basic drugs into acidified compartments such as the extracellular environment. Here we report that hypoxia not only leads to acidification of the tumour microenvironment but also induces endosome hyperacidification. The acidity of the vesicular lumen, together with the alkaline pH of the cytoplasm, gives rise to a strong intracellular pH gradient that drives intravesicular drug trapping and chemoresistance. Endosome hyperacidification is due to the relocalization of the $Na^+/H^+$ exchanger isoform 6 (NHE6) from endosomes to the plasma membrane, an event that involves binding of NHE6 to the activated protein kinase C–receptor for activated C kinase 1 complex. These findings reveal a novel mechanism of hypoxia-induced MDR that involves the aberrant intracellular distribution of NHE6.

[1] Immunology Division, Department of Pediatrics, Faculty of Medicine and Health Sciences, Université de Sherbrooke, 3001, 12th North Avenue, Sherbrooke, Québec, Canada J1H 5N4. [2] Department of Medicine, Faculty of Medicine and Health Sciences, Université de Sherbrooke, Sherbrooke, Québec, Canada J1H 5N4. Correspondence and requests for materials should be addressed to C.M.D. (email: claire.dubois@usherbrooke.ca).

A major challenge in treating cancer is resistance to therapy[1]. A mainstay of therapy for the management of many cancers includes chemotherapy regimens based on anthracyclines or anthracycline analogues (doxorubicin (Dox), daunorubicin (Dnr) or mitoxantrone (Mtx))[2–4]. Yet, the response rates are suboptimal and very few effective therapeutic options are currently available to treat patients who failed to respond to anthracycline treatments[3,5,6].

The success of chemotherapy depends on the ability of the drug to accumulate in the cellular compartment where its target is located (e.g. nucleus). Hence, tumours use various mechanisms to escape the deleterious effect of cytotoxic drugs. Among these, primary or acquired multidrug resistance (MDR) remains the primary hurdle to curative cancer therapy. Although drug resistance is most often attributed to genetic alterations, one major factor contributing to drug resistance is the physical tumour microenvironment (pO$_2$ and pH) that has consistently been shown to impede drug accumulation in cancer cells[7–9]. MDR is a complex and multifactorial process with upregulation of cell-surface efflux pumps (such as ATP-binding cassette transporter family of p-glycoprotein and MDR-associated proteins) being the most studied and clinically tested aspect[10]. Cells adapted to a hypoxic and acidic microenvironment in vitro display upregulated activity of p-glycoprotein, which is thought to contribute to drug resistance[11,12]. However, results from clinical trials targeting these transporters have been so far rather disappointing and it is clear that more detailed knowledge about the causes and mechanisms of drug resistance are needed to find new ways to counter MDR[13].

Accumulating evidence indicates that sequestration of anticancer drugs in intracellular vesicles outside their targeted compartments contributes significantly to the MDR phenotype[14,15]. One mechanism involved in this process is the pH-dependent drug partitioning within cells caused by a direct effect of pH gradients on drug distribution[16,17]. This model predicts that weakly basic chemotherapeutic drugs, that include anthracyclines, will concentrate in acidic compartments such as intracellular vesicles where they will be trapped in their protonated, membrane-impermeant, form[8]. Given that most commonly used anticancer drugs have nuclear targets, such sequestration into vesicles will not only result in insufficient drug accumulation at the target site but will also increase drug extrusion through exocytosis[18]. Therefore, pH-dependent alterations in intracellular drug distribution is an important fundamental mechanism associated with drug resistance, but surprisingly little is known about the molecular regulators of this process.

To maintain pH homeostasis, cells utilize an array of acid–base modulators, such as the sodium/proton exchangers (NHEs) that are critical regulators of pH within the cell and the extracellular microenvironment. Nine NHE isoforms have been described in human[19]. Na$^+$/H$^+$ exchanger 1–5 (NHE1–5) are located at the plasma membrane, whereas Na+/H+ exchanger isoform 6 (NHE6) and isoform 9 (NHE9) are associated with sorting and recycling endosomes, and NHE7 and NHE8 with the trans- and mid-trans-Golgi stacks, respectively[20]. By facilitating proton efflux, organellar NHEs are thought to counteract the acidity generated by vacuolar (V)-ATPase, thereby limiting luminal acidification[21]. Altered expression or mutations in the SLC9A6 or SLC9A9 genes encoding NHE6 and NHE9, respectively, have been associated with neurological diseases in human such as the X-linked Christianson syndrome, familial autism and attention deficit hyperactivity disorder[22–24]. Upregulation of NHE9 has been recently linked to tyrosine kinase inhibitors resistance in oesophageal squamous cell carcinoma and glioblastomas as well as tumour growth and chemoresistance in glioblastomas[25,26]. Although these findings highlight the role of NHE9 in various cancers, the potential involvement of NHE6 in cancer progression or treatment, in particular pH-mediated chemoresistance, remains unknown.

Here, we tested the hypothesis that hypoxia promotes therapy resistance to weak base chemotherapeutics through pH disturbance of intravesicular compartments. We have previously reported that hypoxia promoted NHE1 exchanger activity, resulting in a reversal of pH gradient across the plasma membrane[27]. Herein, we present evidence that hypoxia also triggers acidification of the endosomal compartments that leads to exacerbation of the vesicular pH gradient. Mechanistically, hypoxia-induced acidification in endosomal pH is due to mislocalization of the NHE6 exchanger because of its enhanced binding to receptor for activated C kinase 1 (RACK1) through a protein kinase C (PKC)-dependent mechanism. We further show that a peptide encompassing the RACK1–NHE6 interface reverses alterations in endosomal pH and sensitizes in vitro and in vivo cancer cells to weak base chemotherapeutics. We thereby propose a novel mechanism of MDR that involves translocation of NHE6 from endosomes to the plasma membrane with the consequence of pH-dependent resistance to weak base chemotherapeutic drugs.

## Results

**Hypoxia induces pH-dependent Dox resistance.** To assess the role of pH in hypoxia-induced resistance to anthracyclines, the human breast cancer MDA-MB-231 and fibrosarcoma HT-1080 cell lines were incubated in the presence or absence of the commonly used anthracycline drugs, Dox, Dau and the anthra-cycline analogue drug, Mtx, under hypoxic (1% O$_2$) or normoxic (21% O$_2$) conditions for 72 h. Cell viability was then assessed using the 3-(4,5-dimethylthiazol-2-yl)-2,5-diphenyltetrazolium bromide (MTT) assay. In MDA-MB-231 cells, hypoxia increased resistance towards the chemotherapeutic drugs by 5.5-, 5.0- and 4.7-fold for Dox, Mtx and Dau, respectively (Table 1 and Supplementary Fig. 1a). Similar changes in cell viability were observed in HT-1080 cells (Table 1 and Supplementary Fig. 1d). By their biochemical properties, anthracyclines and their analo-gues are sensitive to pH and are preferentially localized in acidic environments[17,28]. To assess whether pH alterations would affect hypoxia-induced resistance, the vacuolar pH-alkalizing drug chloroquine (Cq) and the V-ATPase inhibitor, bafilomycin A1 (Baf) were used in the presence of the model anthracycline drug, Dox, in viability assays. Treatment of MDA-MB-231 and HT-1080 cells with vacuolar pH-neutralizing agents prevented drug resistance induced by hypoxia with effective concentration values similar to those observed under normoxic conditions (Table 2 and Supplementary Fig. 1b,c,e,f). These results were confirmed using direct cell counting of viable cells using the Trypan blue exclusion method (Supplementary Fig. 1g–j). Therefore, hypoxia promotes cell resistance to anthracycline drugs that can be prevented by pH neutralization.

**Hypoxia promotes Dox sequestration within endosomes.** Given the resensitizing impact of pH neutralizers on hypoxia-induced Dox resistance and the fact that intracellular anthracycline distribution was shown to depend on pH gradients[29], we assessed whether the cellular distribution of Dox was influenced by hypoxia. We took advantage of the native red fluorescence property of Dox that allows monitoring of drug partitioning by fluorescence confocal microscopy in living cells. Consistent with the fact that anthracyclines target topoisomerase II, we observed a predominant accumulation of Dox in the nucleus of normoxic MDA-MB-231 or HT-1080 cells (Fig. 1a,b). When cells were incubated under hypoxic conditions, Dox localization in the nucleus was significantly decreased, whereas fluorescence was increased within perinuclear vesicles. Costaining with Alexa Fluor

**Table 1 | IC$_{50}$ values of anthracyclines for cancer cell lines in normoxia or hypoxia.**

| Cell line | MDA-MB-231 | | | | HT-1080 | | | |
|---|---|---|---|---|---|---|---|---|
| Treatment | 21% O$_2$ | 1% O$_2$ | Fold change | *P*-value | 21% O$_2$ | 1% O$_2$ | Fold change | *P*-value |
| Dox | 125.3 ± 2.7 | 688.1 ± 28.4 | 5.5 | < 0.0001 | 73.4 ± 3.9 | 301.2 ± 30.9 | 4.1 | < 0.0001 |
| Mtx | 151.7 ± 9.4 | 752.2 ± 37.5 | 5.0 | < 0.0001 | 93.5 ± 7.6 | 451.2 ± 23.8 | 4.9 | < 0.0001 |
| Dau | 198.8 ± 4.8 | 921.3 ± 69.3 | 4.7 | 0.029 | 101.9 ± 30.5 | 813.8 ± 29.3 | 8.1 | < 0.0001 |

Dau, daunorubicin; Dox, doxorubicin; Mtx, mitoxantrone; IC$_{50}$, half-maximal inhibitory concentration.
Data are presented as the mean (nM) ± standard deviation. Fold change was calculated as the drug IC$_{50}$ of cells exposed to 1% O$_2$ to the IC$_{50}$ of cells exposed to 21% O$_2$. $n = 3$–5 independent experiments with three replicates in each experiment. *P* values were determined using unpaired *t*-test with Welch correction.

**Table 2 | IC$_{50}$ values of doxorubicin for cancer cell lines treated with neutralizing agents in normoxia or hypoxia.**

| Cell line | MDA-MB-231 | | | | HT-1080 | | | |
|---|---|---|---|---|---|---|---|---|
| Treatment | 21% O$_2$ | 1% O$_2$ | Fold change | *P*-value | 21% O$_2$ | 1% O$_2$ | Fold change | *P*-value |
| Dox | 167.7 ± 30.4 | 852.3 ± 42.7 | 5.1 | < 0.0001 | 113.4 ± 11.9 | 447.7 ± 22.1 | 4.0 | < 0.0001 |
| Dox + Baf (100 nM) | 188.1 ± 17.7 | 86.9 ± 2.9 | − 0.4 | 0.0012 | 40.8 ± 3.1 | 73.5 ± 12.9 | 1.8 | 0.042 |
| Dox + Cq (10 µM) | 46.7 ± 7.2 | 51.4 ± 4.4 | 1.1 | 0.317 | 102.4 ± 8.7 | 97.3 ± 9.2 | − 0.9 | 0.52 |

Baf, bafilomycin; Cq, chloroquine; Dox, doxorubicin; IC$_{50}$, half-maximal inhibitory concentration.
Data are presented at the mean (nM) ± standard deviation. Fold change was calculated as the drug IC$_{50}$ of cells exposed to 1% O$_2$ to the IC$_{50}$ of cells exposed to 21% O$_2$. $n = 3$–5 independent experiments with three replicates in each experiment. *P* values were determined with unpaired *t*-test with Welch's correction.

488-conjugated transferrin (Tfn) indicated that a large proportion of these vesicles corresponded to early and recycling endosomes (Fig. 1a–d). The increase in intravesicular Dox localization was not associated with an increase in total number of endosomes or lysosomes per cell (Fig. 1e,f). Incubation of the cells in the presence of the pH-neutralizing agents (Cq and Baf) abolished hypoxia-induced Dox sequestration within endosomes and restored the preferential accumulation of the drug in the nucleus (Fig. 1g–j). We concluded that hypoxia-induced resistance to Dox is related to drug sequestration within endosomal/recycling vesicles in a pH-dependent manner.

**Dox sequestration is linked to endosome hyperacidification.** The finding that the increased endosomal sequestration of Dox was pH dependent led us to investigate the influence of hypoxia on the pH of endosomal compartments. To do so, we designed an optimized dual ratiometric approach that uses the pH-sensing probes 8-hydroxypyrene-1,3,6-trisulfonic acid (HPTS) and seminaphtharhodafluor-1 (SNARF-1) to measure endosomal–lysosomal pH (pH$_{e/l}$) and cytosolic pH (pH$_c$), respectively, in living cells[30]. After a 16 h incubation of MDA-MB-231 or HT-1080 cells with HPTS, the dye was taken up by pinocytosis, and compartmentalized inside endosomal and lysosomal vesicles with a preferential localization within endosomes (Fig. 2a,b). This was shown by the percentage of colocalization with the endosome marker Tfn ($\sim$60–70%) compared to the lysosomal dye lysotracker ($\sim$20–30%) (Fig. 2c,d). To assess the role of hypoxia in pH$_{e/l}$, we performed a time-course study of pH$_{e/l}$ measurement in cells incubated under normoxic or hypoxic conditions. Data showed that under normoxic conditions, the pH$_{e/l}$ in MDA-MB-231 and HT-1080 cells were mildly acidic with pH values of 6.38 ± 0.17 and 6.62 ± 0.30, respectively. Hypoxia induced a hyperacidification of the vesicles with a decrease in pH$_{e/l}$ reaching 5.77 ± 0.15 for MDA-MB-231 cells and 5.80 ± 0.10 for HT-1080 cells (Fig. 2e,f). To further detail the effects of hypoxia on endosomal pH, cells were colabelled with fluorophore-conjugated Tfn and the pH of early and recycling endosomes was determined. Interestingly, Tfn-positive endosomes were significantly more acidic under hypoxic conditions as compared to normoxia, with a pH change of 0.4 and 0.6 pH units for MDA-MB-231 and HT-1080 cells, respectively (Fig. 2g,h). Conversely, and as expected, hypoxia led to an increase in pH$_c$, which was prevented by the NHE1 inhibitor, EIPA (Fig. 2i,j)[27]. Cytosol alkalinization and endosome acidification resulted in an exacerbation of the pH gradient across the endosomal membranes (ΔpH) with a difference of 0.73 and 1.03 pH units for MDA-MB-231 and HT-1080 cells, respectively (Supplementary Table 1). As expected, hypoxia-induced intravesicular acidification was prevented by the use of the pH-neutralizing agents Cq or Baf (Fig. 2k,l). Taken together, the data indicate that hypoxia leads to alterations in intracellular pH homeostasis resulting in an increase of the pH gradient across the endosomal membranes, a finding that is related to the observed Dox compartmentalization within acidified endosomes and Dox resistance.

**NHE6 delocalization triggers endosome hyperacidification.** We next investigated the mechanisms that could account for endosome acidification in hypoxia. A few proteins have been reported to be involved in maintaining endosomal pH homeostasis. These include the V-ATPase, chloride channels and the newly described organellar NHE6, which is predominantly located at early endosomes, and NHE-9, which is associated at recycling and late endosomes[20]. Given that these NHE isoforms are thought to be responsible for removing protons from the endosome lumen, thereby preventing excess acidification due to V-ATPase activity[19], we sought to evaluate their potential contribution to hypoxia-induced endosome acidification. Silencing of NHE6 in HT-1080 or MDA-MB-231 cells resulted in hyperacidification of the intravesicular compartment to a level similar to that induced by hypoxia, whereas depletion of the NHE9 isoform had no significant impact (Fig. 3a–e and Supplementary Fig. 2a,c). In contrast, over-expression of NHE6 increased the pH of intracellular vesicles and blocked the acidification induced by hypoxia (Supplementary Fig. 2e). NHE6 knockdown in normoxic cells was also associated with increased sequestration of Dox within endosomes in HT-1080 ($\sim$20%, shScrambled versus $\sim$45%, shNHE6)

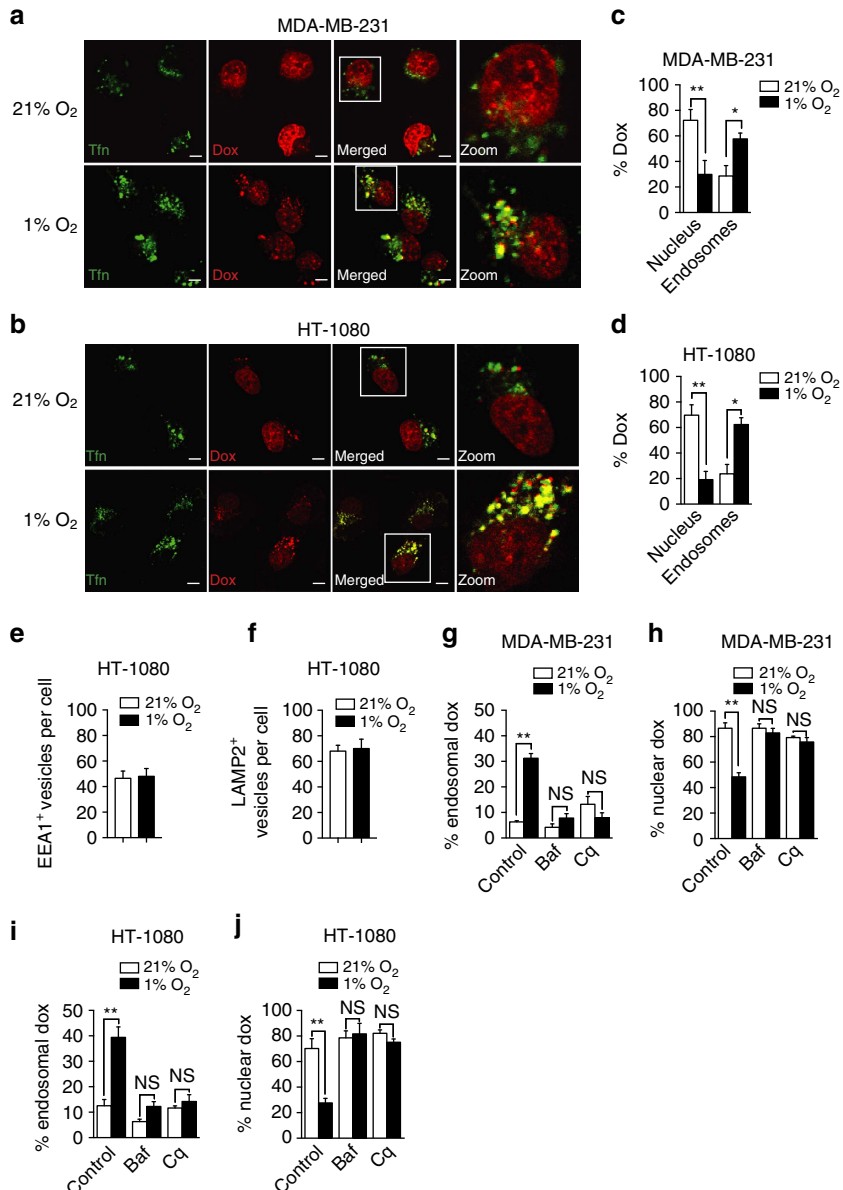

**Figure 1 | Hypoxia triggers Dox sequestration within endosomal compartments.** (**a** and **b**) Representative confocal microscopy images showing Dox fluorescence in (**a**) MDA-MB-231 and (**b**) HT-1080 cells cultured for 4 h in normoxia (21% $O_2$) or hypoxia (1% $O_2$). Endosomal compartments were stained with Alexa Fluor 633-conjugated Tfn. Scale bar, 5 μm, original magnification is ×60. (**c** and **d**) Percentage of Dox fluorescence within Tfn-labelled endosomes or DAPI-stained nucleus in (**c**) MDA-MB-231 and (**d**) HT-1080 cells ($n = 3$ independent experiments with >75 cells per experimental condition). (**e** and **f**) Quantitation of (**e**) EEA1[+] and (**f**) LAMP2[+] per cell in HT-1080 cells cultured for 4 h in normoxia or hypoxia ($n = 3$ independent experiments with >25 cells per experimental condition). (**g**–**j**) Percentage of Dox fluorescence within (**g** and **i**) Tfn-labelled endosomes or (**h** and **j**) DAPI-stained nucleus of MDA-MB-231 and HT-1080 cells cultured under normoxic or hypoxic conditions in the presence or absence of neutralizing agents: Cq (10 μM) or Baf (100 nM) ($n = 3$ independent experiments with >75 cells per experimental condition). Bars represent the mean ± s.e.m. (*$P < 0.05$, **$P < 0.01$, unpaired Student's t-test).

(Fig. 3f,g) or MDA-MB-231 cells (∼10%, shScrambled versus ∼74%, shNHE6) (Supplementary Fig. 2d), an event that correlated with a 2.4-fold increase in cell resistance to Dox reaching half-maximal inhibitory concentration ($IC_{50}$) values similar to the ones observed under hypoxia (Fig. 3h). In contrast, NHE9 silencing did not significantly affect Dox sequestration (Fig. 3g) or cell sensitivity to Dox (Supplementary Table 2). These findings indicate that among the two endosome-located NHE isoforms, only depletion of NHE6 influences Dox sequestration in the endosomal compartment, an observation associated with drug resistance.

Because NHE6 is known to recycle between intravesicular compartments and the plasma membrane and interference with NHE6 trafficking or loss of the protein resulted in hyperacidification of the endosomal compartment[31], we sought to determine whether NHE6 compartmentalization was altered by hypoxia. In normoxic cells, NHE6 strongly colocalized with the early endosomal markers EEA1 (∼62%) and early and recycling endosomal marker Tfn (∼76%), but not with the late endosomal–lysosomal marker Rab7 (Supplementary Fig. 3a,b). Using biotinylated cells to assess levels of NHE6 associated with the plasma membrane, we also observed that a small percentage

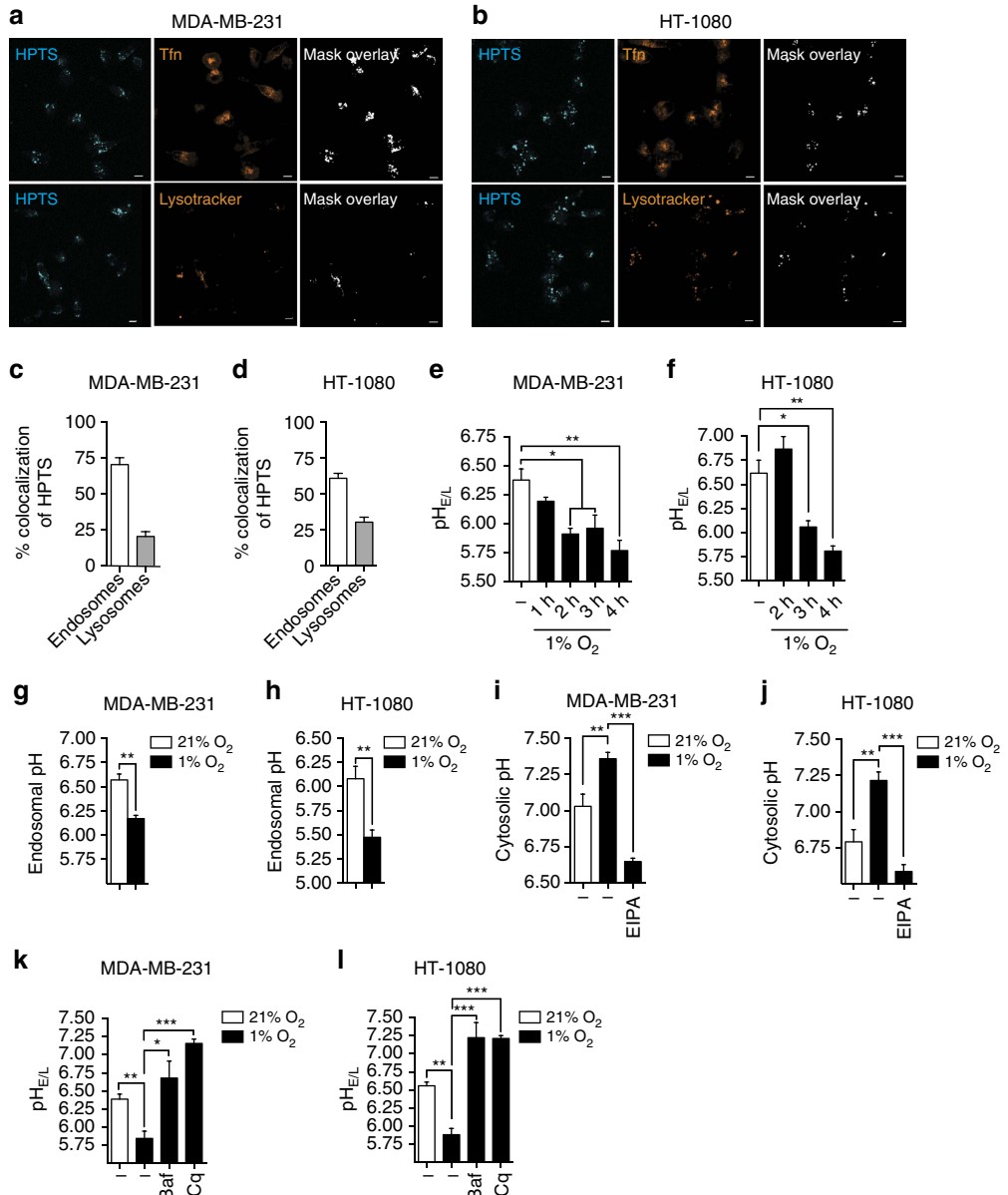

**Figure 2 | Hypoxia promotes endosome hyperacidification.** (**a** and **b**) Representative confocal images of pH-sensitive probe HPTS localization in (**a**) MDA-MB-231 or (**b**) HT-1080 cells labelled with Alexa Fluor 546-conjugated Tfn (endosomes) or Lysotracker (lysosomes). (**c** and **d**) Percentage of colocalization of HPTS with Alexa Fluor 546-conjugated Tfn or Lysotracker showing predominant labelling within endosomes in (**c**) MDA-MB-231 or (**d**) HT-1080 cells ($n = 3$ independent experiments with $>150$ cells per experimental condition). Scale bar, 10 μm and original magnification is $\times 40$. (**e** and **f**) Endosomal–lysosomal pH measured in (**e**) MDA-MB-231 or (**f**) HT-1080 cells cultured in normoxia or hypoxia ($n = 3$ independent experiments with $>75$ cells per condition). (**g** and **h**) Endosomal pH (Tfn-positive vesicles) in (**g**) MDA-MB-231 or (**h**) HT1080 cells cultured for 4 h under normoxic or hypoxic conditions ($n = 3$ independent experiments with $>75$ cells per experimental condition). (**i** and **j**) $pH_c$ measured with SNARF-1 pH-sensing probe in (**i**) MDA-MB-231 or (**j**) HT1080 cells cultured for 4 h under normoxic or hypoxic conditions in the presence or absence of EIPA, a selective NHE1 inhibitor (25 μM) ($n = 3$ independent experiments with $>75$ cells per experimental condition). (**k** and **l**) Endosomal–lysosomal pH in (**k**) MDA-MB-231 or (**l**) HT-1080 cells cultured in normoxia or hypoxia for 4 h in the presence or absence of the neutralizing agents, Baf (100 nM) or Cq (10 μM) ($n = 3$–5 independent experiments with $>75$ cells per experimental condition). Bars represent the mean ± s.e.m. (*$P < 0.05$, **$P < 0.01$, ***$P < 0.001$, unpaired Student's $t$-test).

of NHE6 ($\sim 10\%$ in MDA-MB-231 cells; $\sim 9\%$ in HT-1080 cells) was located at the plasma membrane under basal conditions (Fig. 4a–c). These results confirm that under normoxia, NHE6 is mainly localized at early and recycling endosomes with low levels also found at the plasma membrane[32].

Interestingly, following a 4 h incubation period under hypoxic conditions, the proportion of NHE6 at the plasma membrane was increased threefold in MDA-MB-231 cells and fourfold in HT-1080 cells (Fig. 4a–c). This increase was not associated with induction of NHE6 mRNA or protein levels (Fig. 4d–g). In addition, incubation of HT-1080 cells under hypoxic conditions for 24 h resulted in a similar increase in NHE6 distribution to the plasma membrane as compared to the shorter (4 h) incubation time (Supplementary Fig. 4a,b). Re-exposure of the cells to ambient oxygen levels reduced plasma membrane NHE6 levels (Supplementary Fig. 4a,b), suggesting that the

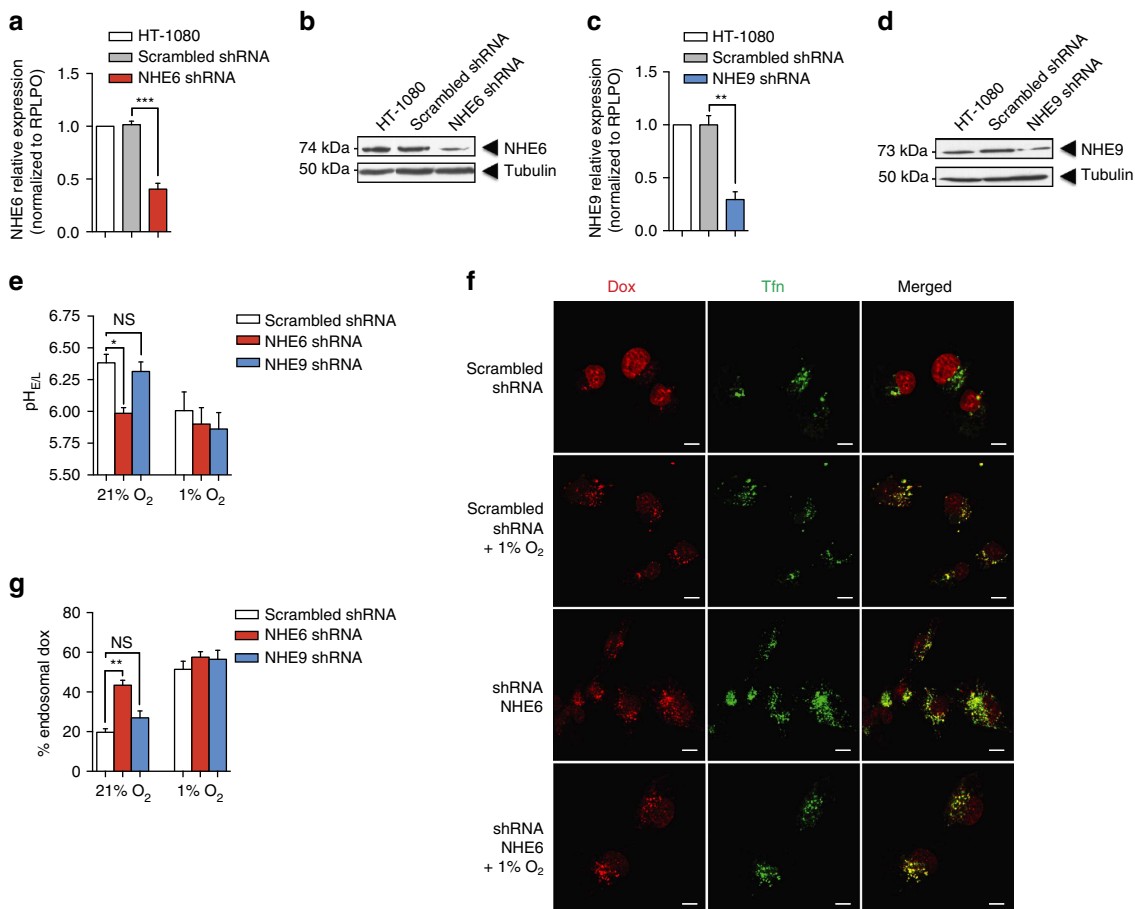

**Figure 3 | NHE6 is a key NHE involved in hypoxia-induced endosome acidification and Dox resistance.** (**a**) mRNA and (**b**) protein levels of NHE6 in HT-1080 cells stably transfected with shRNA directed against NHE6 or scrambled shRNA. (**c**) mRNA and (**d**) protein levels of NHE9 in HT-1080 cells stably transfected with shRNA against NHE9 or scrambled shRNA. RPLPO and α-tubulin were used as controls for qPCR and western blots, respectively ($n = 4$ independent experiments). (**e**) Endosomal–lysosomal pH in NHE6- or NHE9-knockdown cells cultured under normoxic or hypoxic conditions for 4 h ($n = 3$ independent experiments with $> 75$ cells per experimental condition). (**f** and **g**) Representative confocal images in (**f**) NHE6-knockdown cells and (**g**) quantification of Dox within Tfn-positive endosomes of NHE6- or NHE9-knockdown cells cultured under normoxic or hypoxic conditions for 4 h ($n = 3$ independent experiments with $> 75$ cells per experimental condition). Scale bar, 5μm and original magnification is $\times 60$. Bars represent the mean ± s.e.m. (**$P < 0.01$, ***$P < 0.001$, unpaired Student's *t*-test).

mobilization of NHE6 to the plasma membrane is a reversible event.

Costaining with EEA1 indicated that the increased localization of NHE6 at the plasma membrane was associated with a 1.8- and 2.3-fold reduction in its localization at the EEA1$^+$-endosomal compartment in MDA-MB-231 and HT-1080 cells, respectively (Fig. 4h–j). These data suggest that the effects of hypoxia on Dox sequestration and cell resistance were related to the redistribution of NHE6 from endosomes to the plasma membrane, a mechanism that leads to hyperacidification of the endosomal lumen.

**RACK1–PKC–NHE6 axis regulates Dox resistance in hypoxia.** RACK1 is a scaffold protein that regulates plasma-membrane expression of various receptors and Na$^+$/H$^+$ exchangers and RACK1 has also been reported to be associated with chemoresistance of hepatocellular carcinoma cells[31,33]. These reports prompted us to explore the potential involvement of RACK1 in the plasma membrane accumulation of NHE6. Co-immunoprecipitation assays consistently showed that incubation of HT-1080 cells under hypoxia increased the interaction of RACK1 with NHE6 as early as 1 h after treatment (Fig. 5a).

To address the potential link between RACK1–NHE6 binding and NHE6 accumulation at the plasma membrane, we used biotin-labelled RACK1-depleted (short interfering RNA (siRNA)) cells. RACK1 knockdown resulted in a complete blockade of hypoxia-induced NHE6 relocalization from EEA1$^+$ endosomes to the plasma membrane (Fig. 5b–d). Interestingly, RACK1 depletion also restored normal endosomal pH and prevented the sequestration of Dox within the endosomal compartment (Fig. 5e–g).

RACK1 was initially characterized as an intracellular receptor for activated PKC and PKC association has been shown to promote the interaction with transmembrane receptors such as CFTR, AR and PDE4D5 resulting in modulation of their stability and activity[34–37]. We thus investigated whether PKC was involved in hypoxia-induced NHE6–RACK1 interaction.

To first show that PKC is activated by hypoxia, we used a phospho-(Ser) PKC substrate Ab in western blotting of total cell lysates from cells incubated under hypoxic conditions for various time points. Results indicated that hypoxia rapidly increases the phosphorylation of intracellular PKC substrates in HT-1080 and MDA-MB-231 cells with maximal effects observed at 1 and 2 h (Fig. 6a,b). Co-immunoprecipitation assays using HT-1080 cells resulted in detection of PKC in the RACK1 immunoprecipitate

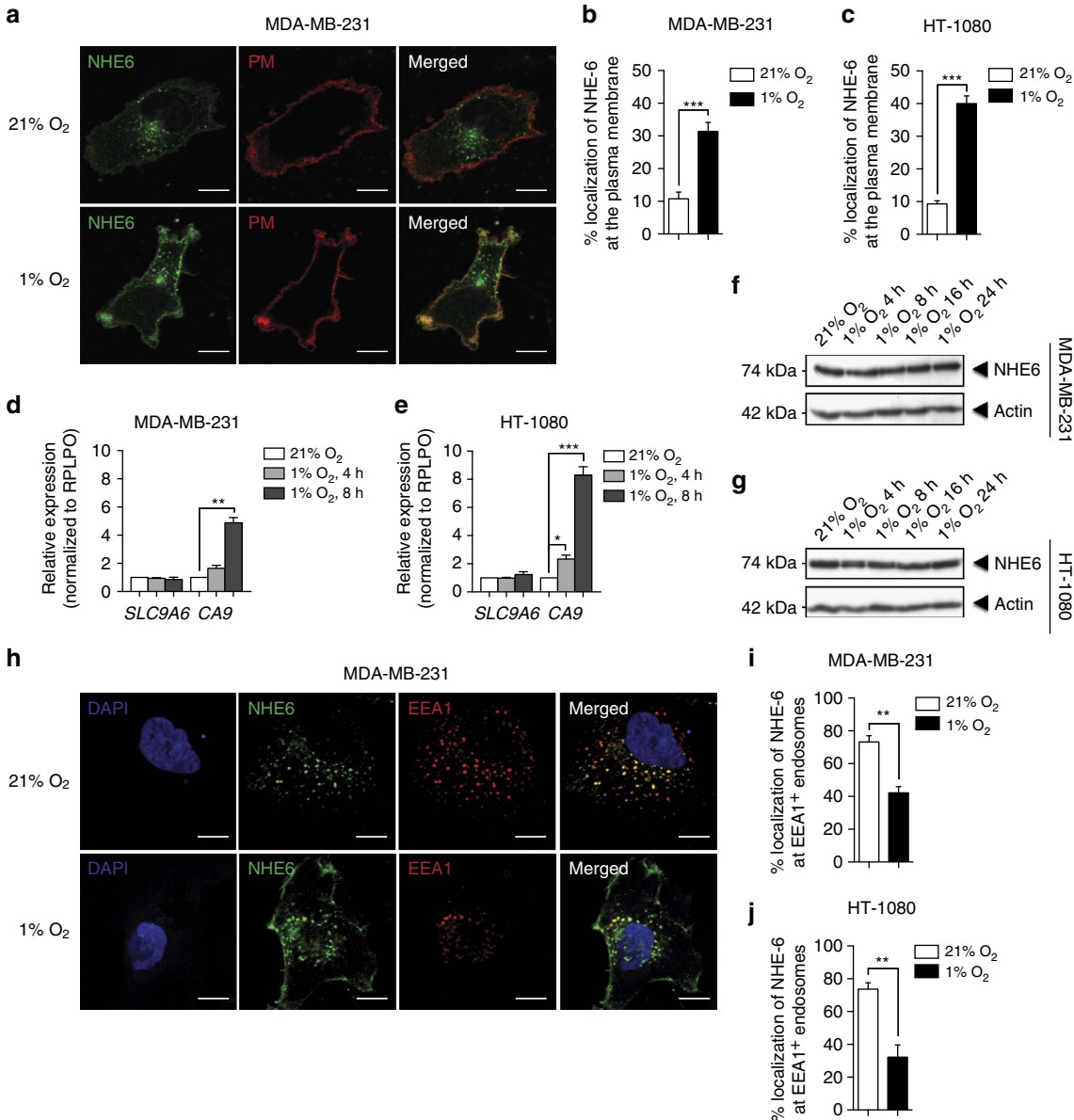

**Figure 4 | NHE6 is relocalized to the plasma membrane (PM) in hypoxia leading to endosomal acidification.** (**a–f**) MDA-MB-231 or HT-1080 cells stably transfected with NHE6-GFP were incubated under normoxic or hypoxic conditions for 4 h. (**a**) Representative confocal images of NHE6 localization in MDA-MB-231 cells. PM was stained by cell-surface biotinylation. Scale bar, 10 μm and original magnification is × 60. (**b** and **c**) Percentage of NHE6 at the (**b**) PM of MDA-MB-231 or (**c**) HT-1080 cells (n = 4 independent experiments with > 100 cells per experimental condition). (**d–g**) mRNA and protein levels of NHE6 and cabonic anydrase 9 (CA9) in (**d** and **f**) MDA-MB-231 and (**e** and **g**) HT-1080 cells cultured in normoxia or hypoxia for the indicated time. CA9 mRNA expression was used as a positive hypoxia-responsive gene. RPLPO and β-actin were used as controls for qPCR and western blots, respectively (n = 3 independent experiments). (**h**) Representative confocal images of colocalization of NHE6 with early endosome marker EEA1 in MDA-MB-231 cells. Scale bar, 10 μm, original magnification is × 60. (**i** and **j**) Percentages of NHE6 colocalization with EEA1 in (**i**) MDA-MB-231 or (**j**) HT-1080 cells (n = 3 independent experiments with > 75 cells per experimental condition). Bars represent the mean ± s.e.m. (**P < 0.01, unpaired Student's t-test).

only in cells treated with the PKC activator phorbol-12,13-dibutyrate (PDBu) or cells incubated under hypoxic conditions, suggesting that RACK1 interacts with activated PKC in hypoxia (Fig. 6c). Treatment of the cells with PDBu promoted the association of NHE6 with RACK1 (Fig. 6d) and the relocalization of NHE6 from endosomes to the plasma membrane under normoxia (Fig. 6e). In contrast, inhibition of PKC with GF109203X[38,39] blocked hypoxia-induced NHE6 relocalization (Fig. 6f), indicating that the plasma membrane mobilization of NHE6 in hypoxia is associated with PKC activation. Consistent with this interpretation, PKC inhibition allowed partial recovery of normal pH values under hypoxic conditions (Fig. 6g) and promoted the accumulation of Dox in the nuclear compartment (Fig. 6h). These results suggest that the increased binding of NHE6 to RACK1 occurs through a PKC-dependent mechanism and that this event regulates the delocalization of NHE6 under hypoxia resulting in changes in endosomal pH and drug sequestration.

**NHE6–RACK1 blockade partially restores Dox sensitivity.** The middle portion (positions 527–588) of the NHE6 cytoplasmic tail has previously been reported to interact with RACK1 (ref. 31).

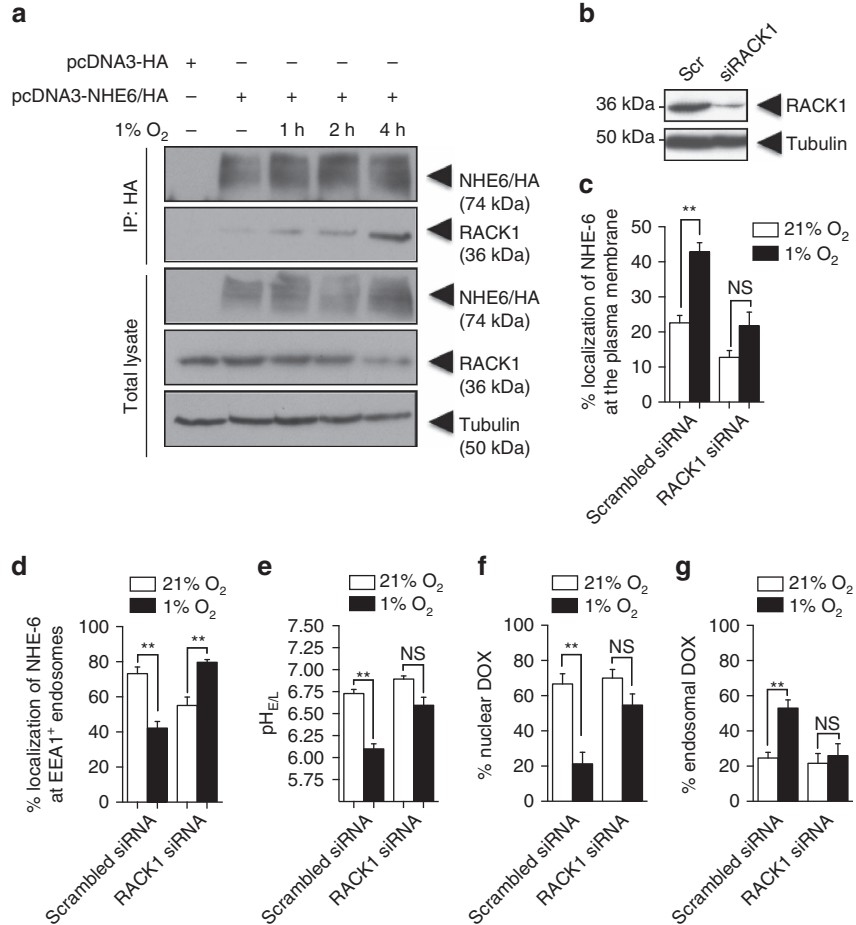

**Figure 5 | Involvement of RACK1 in NHE6 relocalization to the plasma membrane.** (**a**) Co-immunoprecipitation of endogenous RACK1 with HA-tagged NHE6 from transiently transfected HT-1080 cells incubated under normoxia or hypoxia for the indicated times. Data represent 5% of the total cell extract used for each immunoprecipitation. (**b**) Immunoblot analysis of HT-1080 cell lysates 48 h after transfection with non-targeting siRNA or RACK1-specific siRNA. α-tubulin was used as a loading control. The blots shown in (**a**) and (**b**) are representative of four and three independent experiments, respectively. (**c** and **d**) Quantification of NHE6-GFP at the (**c**) plasma membrane and at (**d**) EEA1-positive endosomes in HT-1080 cells transfected with scrambled siRNA or RACK1-specific siRNA ($n = 3$–4 independent experiments and $> 75$ cells per experimental condition). (**e**) Endosomal–lysosomal pH measured in HT-1080 cells in the presence of non-targeting siRNA or RACK1-specific siRNA ($n = 3$ independent experiments with $> 75$ cells per experimental condition). (**f** and **g**) Percentage of Dox fluorescence within (**f**) DAPI-stained nucleus and (**g**) Tfn-labelled endosomes of HT-1080 cells ($n = 3$–4 independent experiments and $> 75$ cells per experimental condition). Bars represent the mean ± s.e.m. (**$P < 0.01$, unpaired Student's $t$-test).

Based on this information, we generated stable HT-1080 cells expressing the NHE6[527–588] fragment or a scrambled counterpart. Results from co-immunoprecipitation assays indicated that ectopic expression of the NHE6[527–588] sequence, but not the control peptide sequence, prevented hypoxia-induced interaction of NHE6 with RACK1 (Fig. 7a). Furthermore, expression of NHE6[527–588] in cells greatly impaired hypoxia-induced NHE6 relocalization from endosomes to the plasma membrane (Fig. 7b,c and Supplementary Fig. 5), intravesicular hyperacidification (Fig. 7d), and Dox accumulation within endosomes (Fig. 7f). In addition, results from cell viability assays indicated that overexpression of the NHE6[527–588] fragment did not change the sensitivity of the cells to Dox under normoxic conditions but a 2.8-fold increase in drug sensitivity was observed in cells exposed to hypoxia (Supplementary Table 3). Taken together, these results demonstrate the usefulness of the NHE6[527–588] sequence in blocking hypoxia-induced NHE6–RACK1 interaction and the ensuing molecular events. They also provide further evidence to support the role of NHE6–RACK1 interactions in hypoxia-regulated Dox resistance in cancer cells.

Our *in vitro* studies indicated that the interaction between RACK1 and NHE6 contributes to intrinsic Dox resistance in

hypoxic cancer cells. To determine whether these observations were of relevance to tumour therapy *in vivo*, we used an *ex ovo* chorioallantoic membrane (CAM) xenograft model in live chicken embryos[40]. MDA-MB-231 or HT-1080 cells, or cells expressing either the NHE6[527–588] competing fragment or its scrambled counterpart, were inoculated onto the CAM of 9-day-old chick embryos. After allowing 3 days for cells to engraft, tumours were treated with Dox or PBS. After 6 days, chick embryos were killed and tumours were removed for analysis (Fig. 8a). To first ascertain that tumours implanted in CAM developed hypoxia, total mRNA was extracted from tumours generated by untransfected HT-1080 cells and gene expression of the hypoxic markers CA9, GLUT1 and MCT4 was measured by quantitative PCR (qPCR). Compared to HT-1080 cells cultured under normoxic conditions, tumour xenografts showed a 5.6-, 3.9- and 4.2-fold increase in CA9, GLUT1 and MCT4 gene expression, respectively (Fig. 8b). In addition, CA9 immunostaining in frozen tumour sections generally colocated with regions of binding of the hypoxic marker, pimonidazole (Fig. 8c), indicating the presence of hypoxic areas in xenograft tumours developed in the CAM assay. Of note, CA9 displayed

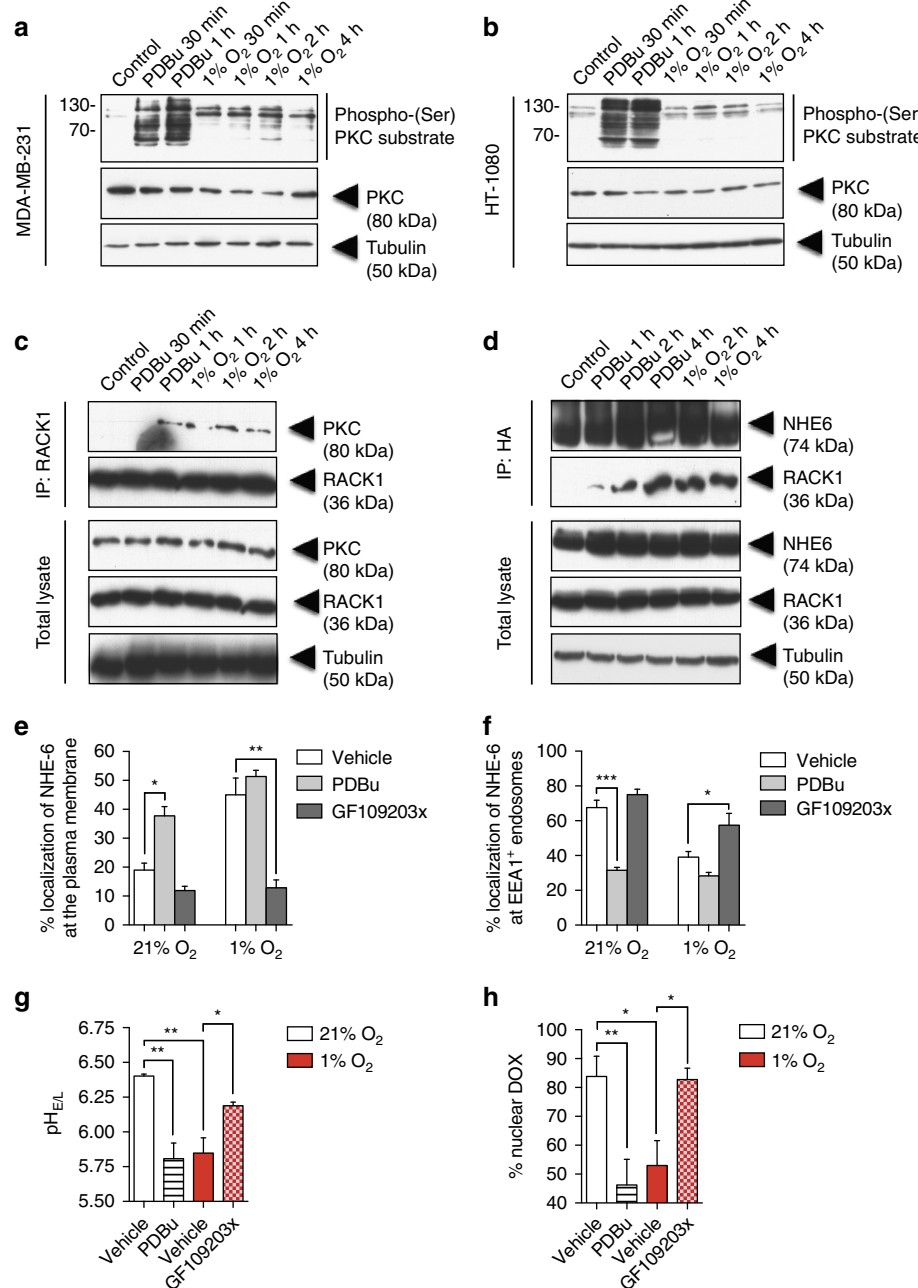

**Figure 6 | NHE6 relocalization to the plasma membrane is dependent on PKC.** (**a**) MDA-MB-231 and (**b**) HT-1080 cells were incubated in normoxia or hypoxia in the presence or absence of the PKC activator PDBu (100 nM). Cell lysates were analysed by western blotting using a phospho-(Ser) PKC substrate antibody. α-Tubulin was used as a loading control. The immunoblot shown is representative of three independent experiments. (**c–f**) HT-1080 cells were cultured for 4 h under 21% $O_2$ or 1% $O_2$ in the presence or absence of PDBu (100 nM), PKC inhibitor GF-109203x (200 nM) or vehicle (dimethylsulfoxide). Co-immunoprecipitation of endogenous (**c**) RACK1-PKC or (**d**) NHE6–RACK1 complex in HT-1080 cells. Data represent 5% of the total cell extract used for each immunoprecipitation. (**e** and **f**) Quantification of NHE6 at the (**e**) plasma membrane and at (**f**) endosomes ($n = 3$–5 independent experiments with $> 80$ cells per experimental condition). (**g**) Endosomal–lysosomal pH measurements ($n = 4$ independent experiments with $> 80$ cells per experimental condition). (**h**) Quantification of Dox within the nucleus ($n = 5$ independent experiments with $> 125$ cells per experimental condition). Bars represent the mean ± s.e.m. (*$P < 0.05$, **$P < 0.01$, ***$P < 0.001$, unpaired Student's $t$-test).

greater areas of staining, consistent with earlier reports showing differences between the pO2 dependency of 2-nitroimidazole binding and CA9 protein expression[41–43]. Dose–response Dox treatment indicated that tumour growth was significantly reduced in HT-1080 and MDA-MB-231 cells using 2 and 5 μM of Dox, respectively (Fig. 8d,e). At lower concentrations (0.5 μM for HT-1080 cells and 1 μM for MDA-MB-231 cells), tumour growth

was unaffected. Treatment of NHE6[527–588]-overexpressing HT-1080 or MDA-MB-231 tumour xenografts with these suboptimal concentrations of Dox lead to a small but significant decrease in tumour volume compared to xenografts overexpressing a control peptide (Fig. 8f–h). These observations indicate that blockade of NHE6–RACK1 interaction improved sensitivity of solid tumours to Dox treatment.

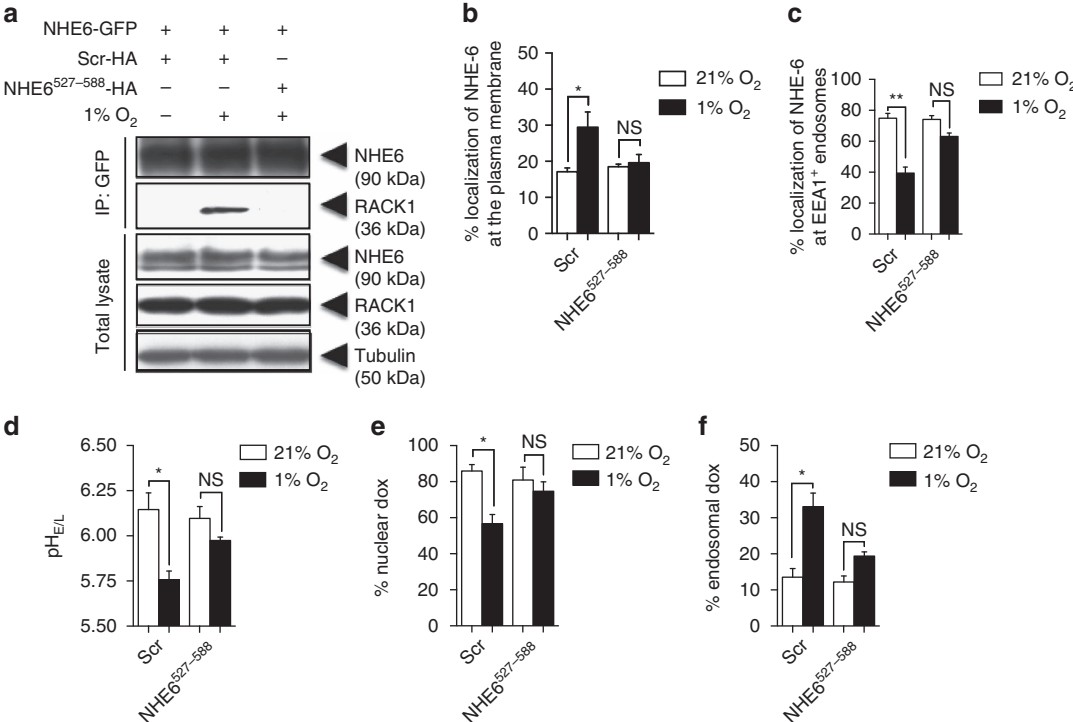

**Figure 7 | Blockade of NHE6–RACK1 interaction prevents NHE6 redistribution to the plasma membrane.** HT-1080 cells stably transfected with NHE6-GFP and transiently transfected with a plasmid encoding the NHE6[527–588] peptide or scrambled peptide were incubated under 1% $O_2$ or 21% $O_2$ for (**a–f**) 4 h or (**g**) 72 h. (**a**) Co-immunoprecipitation of endogenous RACK1 with GFP-tagged NHE6. Data represent 5% of the total cell extract used for each immunoprecipitation. Data are representative of at least four independent experiments. (**b** and **c**) Quantification of NHE6-GFP at the (**b**) plasma membrane and at (**c**) EEA1-positive endosomes ($n = 5$ independent experiments with $> 125$ cells per experimental condition). (**d**) Endosomal–lysosomal pH ($n = 5$ independent experiments with $> 100$ cells per experimental condition). (**e** and **f**) Percentage of Dox within (**e**) DAPI-stained nucleus and (**f**) Tfn-labelled endosomes ($n = 4$ independent experiments with $> 100$ cells per experimental condition). Bars represent the mean ± s.e.m. (*$P < 0.05$, **$P < 0.01$, unpaired Student's $t$-test).

## Discussion

The ability of hypoxic tumour cells to escape chemotherapy through a mechanism that involves pH-dependent drug partitioning within the acidified extracellular microenvironment has been known for several decades and various inhibitors of plasma membrane proton pumps and exchangers are currently being evaluated to sensitize human solid tumours to anticancer drugs[44–46]. Our results uncovered a novel and complementary mechanism through which hypoxia induces drug resistance in cancer cells by regulating pH-dependent drug partitioning within intracellular compartments. We showed that the endosomal NHE6 exchanger induced weak base chemotherapeutic drug resistance when it is relocated to the plasma membrane of hypoxic cancer cells. Using a blocking peptide approach, we were able to prevent the plasma membrane mobilization of NHE6, by reducing its binding to the scaffold RACK1, leading to the reversal of endosome hyperacidification and cancer cell sensitization to Dox.

A predominant hallmark of hypoxic tumours is the acidification of the extracellular microenvironment and the alkalinization of the tumour cells because of the pathological regulation of ion dynamic across the plasma membrane. This resulted in an acid-outside plasmalemmal pH gradient (reversed pH gradient) mostly driven by the increased expression and/or activation of plasma-membrane ion exchangers and transporters such as the NHE1, carbonic anhydrase-9 and monocarboxylate transporters MCT1 and MCT4 (refs 47–50). Such plasmalemmal pH gradient is widely known to act as a physical barrier for cellular uptake of the most currently used weak base chemotherapeutics such as anthracyclines and vinca

alkaloids. Herein, incubation of HT-1080 cells in media of different pHs indicates that extracellular acidification indeed reduces the uptake of the anthracycline drug Dox as observed by the decrease in Dox fluorescence intensity of the nucleus (Supplementary Fig. 6). Several strategies are now being developed to exploit this pH difference for the treatment of cancer[8,16,17,29,51–53]. Among these, the use of proton transport inhibitors such as cariporide and other potent NHE1 inhibitors of the amiloride or non-amiloride series, alone or in combination with conventional chemotherapy, have been proposed to be a promising avenue for improving cancer treatment[46,54]. Some efficacy has been demonstrated with these inhibitors in cellular and preclinical models, although with mixed results[55–57]. Herein, we provide evidence that in addition to acidification of the extracellular microenvironment, a brief exposure of cancer cells to hypoxic conditions triggered hyperacidification of the endosomal compartment. The acidity of the endosomal lumen, together with alkalinization of the cytosol, triggered by proton efflux across the plasma membrane, resulted in a strong vesicular pH gradient that promoted intravesicular accumulation of weakly basic drugs and cell resistance. From the standpoint of the pH-dependent drug partitioning model, our findings mean that current therapies aimed at blocking proton efflux across the plasma membrane of hypoxic cancer cells should increase the intracellular levels of weak base drugs, which will be readily trapped and sequestered into acidified endosomal compartments, hence compromising therapeutic efficiency.

Another therapeutic strategy that could lead to similar intracellular drug accumulation and trapping is the blockade of MDR protein activity. Various inhibitors of MDR proteins are

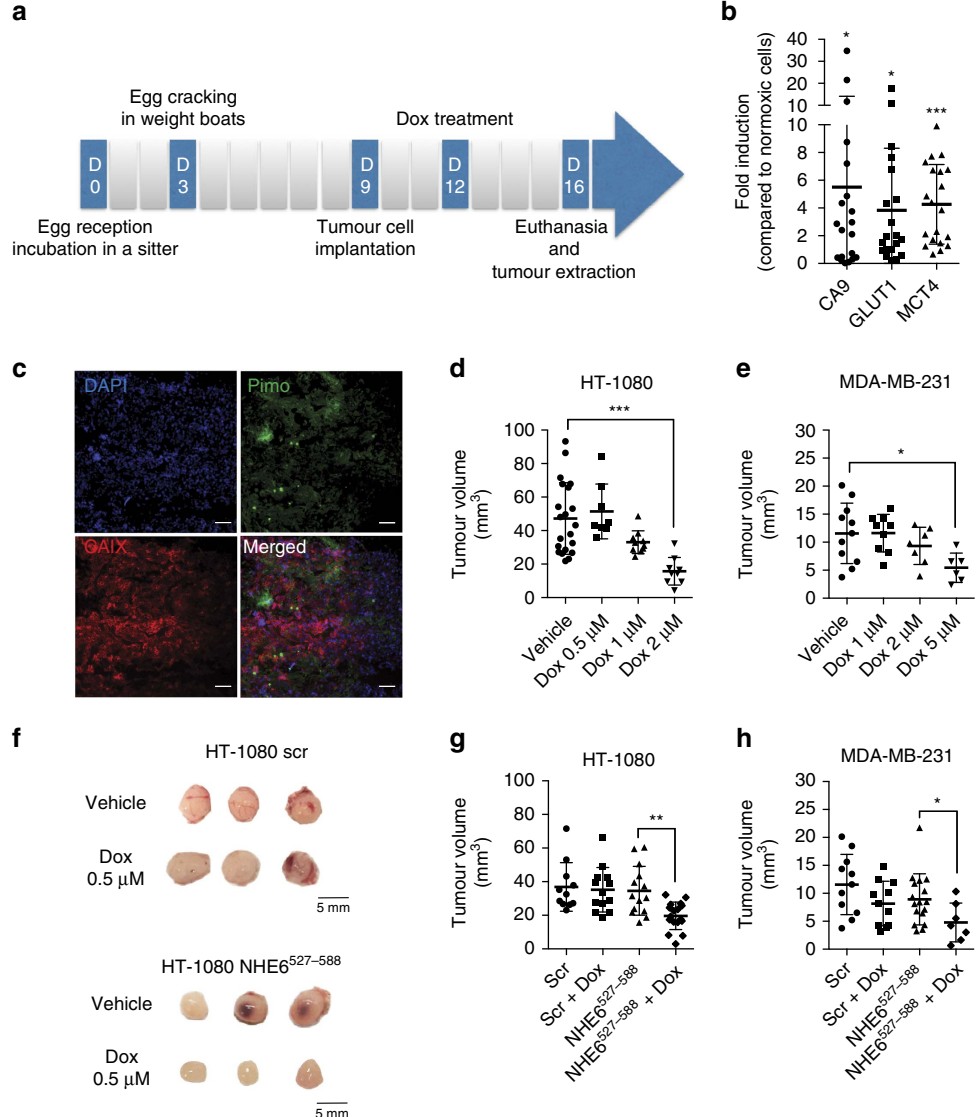

**Figure 8 | Blockade of NHE6–RACK1 interaction minimizes Dox resistance in a chorioallantoic membrane xenograft assay. (a)** Timeline of the human tumour cell xenograft assay in the CAM of chick embryos. **(b)** mRNA expression of hypoxic markers CA9, GLUT1 and MCT4 in HT-1080-derived tumours extracted from the chorioallantoic membrane 7 days after implantation. RPLPO was used as loading control ($n = 4$ independent experiments with 5 tumours per experiment). **(c)** Representative staining of HT-1080 tumours extracted from CAM showing hypoxic regions (Pimo$^+$) and hypoxic cells (CAIX$^+$). Nuclei were stained with DAPI. Scale bar, 100 μm and original magnification is × 10. **(d** and **e)** Tumour volumes from **(d)** HT-1080 and **(e)** MDA-MB-231 cells grown onto CAM and treated with various concentrations of Dox ($n = 5$–7 embryos per group). **(f)** Representative images of tumours grown on CAM from HT-1080 cells transfected with scrambled peptide (HT-1080 scr) or NHE6$^{527–588}$ peptide (HT-1080 NHE6$^{527–588}$) and treated with 0.5 μM Dox. **(g** and **h)** Tumour volumes of **(g)** HT-1080 and **(h)** MDA-MB-231 cells grown on CAM in the presence of scrambled sequence (scr) or NHE6$^{527–588}$ sequence and treated with **(g)** 0.5 μM Dox or **(h)** 1 μM Dox ($n = 5$–9 embryos per group). Bars represent the mean ± s.e.m. (*$P < 0.05$, **$P < 0.01$, ***$P < 0.001$, unpaired Student's $t$-test).

currently being evaluated to circumvent drug resistance. Third-generation p-glycoprotein inhibitors such as tariquidar have high potency and specificity, over the first class inhibitors. However, results from clinical trials have shown limited success in restoring Dox sensitivity in women with stage III–IV breast carcinomas[58]. Here we show that among the cell lines used, only HT-1080 cells express detectable levels of p-glycoprotein. Confocal microscopy analysis of p-glycoprotein staining in HT-1080 cell line indicates that the protein does not colocalize with EEA1$^+$ endosomes or LAMP1$^+$ lysosomes under normoxic or hypoxic conditions data (Supplementary Fig. 7a–e), suggesting that this transporter is not a key component of endosomal Dox sequestration under acute hypoxia. Of note, a small but

significant increases in *MDR1* mRNA expression was observed after 8 h incubation in low $O_2$, which is consistent with a recent report showing that hypoxia can rapidly promote p-glycoprotein expression in laryngeal cancer cells[59]. Even though p-glycoprotein is unlikely to play a direct role in endosomal–lysosomal Dox distribution in acute hypoxia, we cannot exclude the potential contribution of p-glycoprotein in Dox distribution under sustained hypoxia as it is the case in solid tumours. Based on data reported here, it would be of interest to investigate whether inhibition of MDR proteins or restoration of plasmalemmal pH gradient, concomitant with manipulation of the endosomal pH gradient, would improve distribution and efficacy of weak base drugs by depriving hypoxic tumours of

both drug resistance strategies. In this context, various ways to disrupt or neutralize intravesicular pH gradients have been proposed. These include photoacoustic disruption or alkalinization of intracellular acidic vesicles with Cq or V-ATPase inhibitors and such strategies have been shown to reverse Dox resistance *in vitro* and *in vivo*[60]. Even though the toxicity of V-ATPase inhibitor used in this study (Baf) prohibits its *in vivo* and clinical applications, our ability to partially reverse drug resistance induced by hypoxia *in vitro* and in the context of solid tumours using blocking peptides that specifically interfere with the molecular events involved in endosome acidification indicates that such mechanism of drug sequestration can be tested therapeutically through precise intervention.

Another important observation of our study is the finding that the mobilization of NHE6 to the plasma membrane is a critical step in the promotion of Dox resistance in hypoxic cells. Although the overexpression of NHE9 in glioblastoma and oesophageal squamous cell carcinoma has previously been shown to be involved in chemoresistance[25,26], in this study knockdown of NHE9 had no significant impact on endosome pH homeostasis, intracellular Dox partitioning or Dox resistance. The discrepancy between our data and those reported previously may include differences in the physicochemical properties of the chemotherapeutic drugs tested and the possibility that NHE9 expression is restricted to a small endosome population, so it may negligibly be involved in pH homeostasis of cells that do not overexpress this isoform.

Whereas the majority of NHE6 is normally distributed in recycling compartments, only a small amount of NHE6 is found at the plasma membrane under basal conditions ($\sim 10\%$)[31,61]. NHE6 has been shown to be translocated to the plasma membrane in adipocytes upon insulin stimulation, or in osteoblasts during bone mineralization[62,63]. However, the detailed mechanisms responsible for the trafficking of NHE6 are not yet understood. Unlike NHE1, little is known about NHE6-interacting proteins. Two proteins, the AT2 receptor and RACK1, have been identified as direct NHE6-binding partners[31,64]. Changes in cellular levels of RACK1 have been shown to affect the plasma membrane localization of NHE6 as well as endosomal pH[31]. In this report, we observed that the interaction of NHE6 with RACK1 and PKC was markedly increased by hypoxia. Such interaction explains the raise of NHE6 at the plasma membrane since RACK1 knockdown, modulation of PKC activity, or interference with RACK1–NHE6 binding greatly affected hypoxia-induced NHE6 plasma membrane mobilization. Modulation of either PKC activity or RACK1–NHE6 interaction also recovered normal endosomal pH, and restored weak base drug sensitivity in hypoxic cells indicating that plasma-membrane NHE6 mobilization represents an important mechanism by which hypoxia drives drug resistance.

In summary, our findings uncovered a mechanism by which hypoxia induced pH-dependent drug partitioning within hyper-acidified endosomal compartments that involves a RACK–PKC-dependent enhanced sequestration of NHE6 at the plasma membrane. Although future studies are required to determine whether this mechanism operates in patient's tumours, our *in vivo* analysis showing weak base drug sensitization using the RACK1–NHE6 competing peptide suggests that the increased endosomal pH gradient occurs in the context of hypoxic solid tumours. Therapies directed against such a tumour microenvironment-driven mechanism may have interesting clinical potential since they should enhance drug toxicity in hypoxic cells, which are the ones related to metastasis and cancer recurrence[9,65], while sparing normal tissues.

## Methods

**Antibodies and reagents.** Antibodies used for immunofluorescence microscopy or western blotting were obtained from commercial sources. The following antibodies were used: rabbit anti-NHE6 (Abcam; 137185), rabbit anti-NHE9 (Abcam; 167157), mouse anti-RACK1 (BD Biosciences; 610177), rabbit anti-RACK1 (Cell Signaling Technology; 5432), mouse anti-tubulin (Sigma-Aldrich; T6199), rabbit anti-actin (Sigma-Aldrich; A5060), mouse monoclonal anti-HA (BioLegend; MMS-101P), mouse anti-EEA1(Santa Cruz Biotechnology; 6415), mouse anti-Rab7 (Santa Cruz Biotechnology; 10767), rabbit phospho-(Ser) PKC substrate (Cell Signaling Technology; 2261), mouse monoclonal PKC (Santa Cruz Biotechnology; 80), mouse monoclonal anti-LAMP2 (Abcam; 25631), rabbit anti-p-glycoprotein (Abcam; 129450). All Alexa Fluor secondary antibodies were acquired from Thermo Fisher Scientific. 4′, 6-diamidino-2-phenylindol dilactate (DAPI), Alexa Fluor-conjugated Tfn and Lysotracker were purchased from Thermo Fisher Scientific. Chemotherapeutic drugs (Dox, Mtx, Dau) were obtained from the local drug dispensary of the Centre Hospitalier Universitaire de Sherbrooke. Cq and Baf were obtained from Sigma-Aldrich. Cq was dissolved in water and Baf in dimethylsulfoxide. Control short hairpin RNA (shRNA) and shRNA directed against human NHE6 or NHE9 were purchased from Sigma-Aldrich. RNA interference directed against RACK1 was purchased from Ambion.

**Cell culture under hypoxic conditions.** HT1080 fibrosarcoma cells (ATCC) were cultured in Eagle's mimimum essential medium and MDA-MB-231 breast cancer cells (ATCC) were cultured in Dulbecco's modified essential medium. Culture media were supplemented with 10% heat-inactivated FBS and $40\,\mu g\,ml^{-1}$ gentamicin. Cells were cultured in a humidified atmosphere at 37 °C with 5% $CO_2$ and 21% $O_2$. For incubation under hypoxic conditions, cells were placed in an In Vivo$_2$ 400 hypoxia workstation (Ruskinn) under a humidified atmosphere of 1% $O_2$ and 5% $CO_2$. All cell lines were routinely tested for mycoplasma using the MycoSEQ Mycoplasma Detection Kit (Thermo Fisher Scientific).

**Plasmid construction and transfection in human cell lines.** peGFP-N3-NHE6 was generously provided by Hiroshi Kanazawa (Osaka University, Japan). pcDNA3-HA/NHE6 was constructed from peGFP-N3-NHE6. For this, the entire NHE6 coding sequence was excised with *Bam*HI and *Eco*RI restriction enzymes and cloned in pcDNA3-HA vector. pcDNA3-HA/NHE6$^{527-588}$ was designed from pcDNA3-HA/NHE6 using the following primers: forward, 5′-ATGCGGAT-CCACCAAAGCAGAGAGTGCTTG-3′ and reverse, 5′-GCATGAATTCTTAAT-CATCATCTTTCAACTGT-3′.

HT-1080 cells were stably transfected with polyethylenimine (MirusBio) and positive cells were selected with Geneticin (G418) at $400\,\mu g\,ml^{-1}$. In the case of MDA-MB-231 cells, plasmids were transfected using Lipofectamine 2000 (Sigma-Aldrich) and G418 was added to cell cultures at a concentration of $2\,mg\,ml^{-1}$.

**Lentiviruses and cell transduction.** pLKO.1-NHE6 and pLKO.1-NHE9 shRNA (Sigma-Aldrich) were co-transfected with ViraPower Lentiviral Packaging Mix (Invitrogen) into HEK293T cells according to the manufacturer's instructions. Viruses were collected and concentrated by ultracentrifugation 72 h after transfection. HT-1080 and MDA-MB-231 cells were infected overnight with viruses and selected with puromycin ($2\,\mu g\,ml^{-1}$) on the third day following transduction.

**Cell viability assay.** Cells ($5 \times 10_3$) were cultured in 96-well plates and preincubated under 21% $O_2$ or 1% $O_2$ for 4 h following the addition of drugs for 72 h. Cell viability was measured using the MTT dye (Life Technologies) according to the manufacturer's instructions. Because hypoxic conditions may affect the reduction of MTT to formazan, incubation of all cell cultures with MTT were performed under normoxic conditions. Each experiment was performed in triplicate and at least three independent experiments were performed. Log-scale dose–response data were plotted on a graph and a three-parameter nonlinear regression was applied to determine IC$_{50}$ values. For calculation of the IC$_{50}$s, both normoxic and hypoxic drug-treated samples were normalized with their untreated counterpart.

Cell viability was also assessed by the Trypan blue exclusion method. Briefly, cells were grown to 80% confluency, trypsinized and plated in triplicate into 6-well plates. Cells were incubated for 24 h following drug treatment. Cells were collected and equal amount of freshly prepared Trypan blue solution was added to the cell suspension. Viable cells were counted with a haemocytometer and the experiment was repeated at least three times. The percentage of viable cells was determined relative to the number of control cells.

**Intracellular localization of Dox and live-cell imaging.** Cells were cultured in 25-mm-diameter glass coverslips (Thermo Fisher Scientific) and incubated under various conditions (as indicated in the figure legends). Cells were then incubated for 2 h in a medium containing Dox ($2\,\mu M$). After the incubation period, cells were washed to remove excess Dox and left in drug-free media for 10 min. In selected samples, Alexa Fluor 633-conjugated Tfn ($25\,\mu g\,ml^{-1}$) was added to the cells and incubated for 20 min to label endosomes. Coverslips were mounted on glass slides and placed on a 37 °C warmed stage of a Olympus Fluoview FV1000

(Olympus, Tokyo, Japan) confocal laser scanning microscope. Dox was excited with a green helium neon laser (543 nm) and emitted fluorescence intensity was measured. Images were acquired on the same day, typically from 10 to 20 cells of similar size from each experimental condition, using identical instrument settings.

**Intracellular pH measurement by confocal microscopy.** The pH-sensing ratiometric dyes SNARF-1 and HPTS (Life Technologies) were used simultaneously for pH measurement of both cytoplasmic (C-SNARF-1) and endosomal–lysosomal (HPTS) compartments as described previously[30]. Briefly, cells were cultured in 35 mm Petri dishes (BD Biosciences) and used at 50% confluence. For endocytic labelling, cells were incubated overnight with HPTS (1 mM), washed, followed by a 20 min incubation with SNARF-1 ($5 \mu M$) to label the cytoplasmic compartment. Living cells were analysed using an Olympus Fluoview FV1000 confocal microscope. Fluorescence emissions of both pH-sensing probes were recorded and subsequently analysed as described[30].

**Cell-surface biotinylation.** Biotinylation steps were performed at 37 °C. Cells were grown on circular 15-mm diameter glass coverslips (Thermo Fisher Scientific) and incubated at 21% or 1% $O_2$. Cells were then washed with PBS containing 1 mM $CaCl_2$ and 0.5 mM $MgCl_2$ and incubated for 5 min with $0.3 \text{ mg ml}^{-1}$ of EZ-Link Sulfo-NHS-SS-Biotin (Pierce) in PBS containing 1 mM $CaCl_2$ and 0.5 mM $MgCl_2$. Unreacted biotin was quenched using 100 mM glycine. Cells were fixed for 10 min at room temperature using 1% paraformaldehyde (PFA) in PBS. Before staining, cells were blocked with 5% of bovine serum albumin (BSA) for 1 h at room temperature. Biotinylated cells were stained with $5 \mu \text{g ml}^{-1}$ Streptavidin Texas Red (Invitrogen) for 1 h at 4 °C. Coverslips were washed and mounted on a microscope slide using Vectashield mounting media (Vector Labs).

**Immunofluorescence.** Cultured cells were fixed with 1% PFA for 10 min at room temperature, permeabilized with saponin (0.05% in PBS) for 20 min and blocked with 2% BSA in PBS for 30 min. Cells were then incubated with the appropriate primary and secondary antibodies as follows: anti-EEA1 (1/500), anti-Rab7 (1/500) and fluorophore-conjugated secondary antibodies (1/1,000). Images were recorded using an Olympus Fluoview FV1000 confocal microscope using a $\times 63$ oil immersion objective.

**Immunoprecipitation and western blotting.** Cells were lysed on ice in NP-40-containing buffer (50 mM Tris-HCl (pH 8.0), 150 mM NaCl, 1% NP-40, 5 mM EDTA, phosphatase and protease inhibitors). Cell lysates were centrifuged at 13,000 r.p.m. at 4 °C and protein concentration was determined using the BCA protein assay (Thermo Fisher Scientific). After a preclearing step with protein A/G-agarose beads (GE Healthcare), protein complexes were immunoprecipitated overnight at 4 °C. Samples were incubated in SDS-loading buffer for 30 min at room temperature to avoid transmembrane protein aggregates upon heating in a boiling water bath. Proteins were separated by SDS–polyacrylamide gel electrophoresis electrophoresis, transferred to a polyvinylidene difluoride membrane and immunoblotting was performed as described, using anti-NHE6 (1:1,000), anti-NHE9 (1:1,000) anti-RACK1 (1:2,000), anti-tubulin (1:5,000), anti-actin (1:5,000) anti-HA (1:1,000), anti-PKC (1:200), anti-phospho PKC substrate (1:1,000), anti-LAMP2 (1:200), anti-P-glycoprotein (1:200) and horseradish peroxidase-conjugated secondary antibodies (1/10,000)[66]. Uncropped scans of the most important western blots were shown in Supplementary Fig. 8.

**RNA isolation and qPCR analysis.** Total cellular RNA was isolated using the TRI-Reagent protocol (Invitrogen) according to the supplier's protocol. Quantitative real-time PCR was performed using the SYBR Green qPCR Mastermix (Biotool) and a Rotor-Gene 3000 Instrument (Corbett Research). Each reaction was run in duplicates and values were normalized against the RPLPO housekeeping gene. Double delta Ct method was used to determine relative gene expression: $\Delta\Delta Ct = (Ct_{treated} - Ct_{untreated})_{\text{gene of interest}} - (Ct_{treated} - Ct_{untreated})_{\text{housekeeping gene}}$. Fold changes were calculated using the equation: expression fold change $= 2^{-\Delta\Delta Ct}$.

**Chorioallantoic membrane assay.** Fertilized eggs from white leghorn chicken were obtained from Public Health Agency of Canada (Nepean, ON, Canada). Ethics approval was obtained from the Ethics Committee on Animal Research of the University of Sherbrooke and all experimental procedures involving embryos were conducted in accordance with the regulations of the Canadian Council on Animal Care. Eggs were incubated in an Ova-Easy egg incubator (Brinsea) at 37 °C with 60% humidity. At day 3, eggs were cracked as described[67]. At day 9, HT-1080 and MDA-MB-231 cell suspensions ($1 \times 10^6$ and $2 \times 10^6$ cells, respectively) were mixed (1:1) with growth factor reduced Matrigel (BD Biosciences) in a total volume of $20 \mu l$. Cell grafts were placed on top of the CAM and eggs were returned to the incubator for 96 h until day 13 ($n > 6$ chick embryos per cell line). When mentioned, $50 \mu l$ of Dox were added topically to the formed tumours (day 13 after tumour cell grafting). At day 16, chick embryos were killed by decapitation. Tumours were removed and tumour volumes were calculated using the formula: ($Dd^2/3$). For RNA quantification, tumours were immediately

snap frozen in liquid nitrogen and kept at $-80$ °C until RNA extraction using the Trizol reagent.

**Immunohistochemistry.** Tumour hypoxia was determined by intravenous injection of 4 mg pimonidazole hydrochloride (Hydroxyprobe, Hypoxyprobe-1 Kit) in $50 \mu l$ solution for 30 min before the tumour was collected. Tumours removed from CAM were placed directly in the cryopreservative embedding media OCT compoumd (Tissue Tek) and immediately frozen in a mixture of isopentane and carbonic ice. Sections of $5 \mu m$ thickness were fixed with PFA 4% for 10 min at 4 °C. Blocking and staining were performed in BSA 2%, 0.2% Triton X-100 and supplemented with 10% of goat serum. Tumour sections were double stained for pimonidazole in combination with carbonic anhydrase IX (1/50). Pimonidazole was detected with mouse antibody (Hypoxyprobe, 1:200) and goat anti-mouse $IgG\gamma1$-Alexa Fluor 488 (Invitrogen).

**Statistical analysis.** The GraphPad software was used for statistical analysis. Paired or unpaired Student's $t$-test were used to assess statistical significance, which was set at $P$ value $< 0.05$.

**Data availability.** All data generated or analysed during this study are included in this published article (and its Supplementary Information files).

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

## Acknowledgements

We thank Dr Gilles Dupuis and Karine Brochu-Gaudreau for critical reading of the manuscript and helpful comments. We also thank Dr Léonid Volkov for expert assistance with confocal microscopy. This work was supported by Canadian Institutes for Health Research (CIHR) Grant MOP-126173 and Natural Sciences and Engineering Research Council Discovery Grant RGPIN-2016-03928 (to D.C.M.). D.C.M. is a member of the Fonds de la Recherche en Santé du Québec-funded Centre de Recherche du Centre Hospitalier Universitaire de Sherbrooke. K.H. is recipient of a scholarship from CIHR.

## Author contributions

F.L. performed the majority of the experiments and interpretation of the data. P.P.P. performed plasma membrane biotinylation experiments. R.L.R. performed immunoprecipitation and viability assays. D.A. performed NHE6 cellular localization experiments. K.H. contributed to CAM assays. J.M.L. participated to CAM assays and performed qPCR experiments. S.R. performed mutagenesis experiments and engineered plasmid constructs. F.L. and D.C.M. planned the experiments and wrote the manuscript. D.C.M. supervised the study. All authors contributed to interpretation and discussion of the results and approved the final version of the manuscript.

# ARTICLE

## Additional information

**Competing interests:** The authors declare no competing financial interests.

