## [Peer Review File · Nature Communications]

Reviewers' comments:

Reviewer #1 (Remarks to the Author):

Nature Communications 1108840 Commentary

Re. Hypoxia-induced mobilisation of NHE6 to the plasma membrane by Fabrice et al.

This paper shows that mild hypoxia induces mobilisation of the sodium-hydrogen exchanger NHE6 from endosomal compartment to the plasma membrane, dependent on PKC and markedly affecting endosomal pH. The translocation to the plasma membrane results in hyper acidification of early endosomes, which can then trap basic drugs such as doxorubicin, causing drug resistance.

These conventional anticancer drugs, topoisomerase II inhibitors, remain an important contribution to the treatment of many solid tumours, in spite of recent advances in non-chemotherapy approaches. It has long been known that drug resistance is mediated for basic drugs by an acidic microenvironment and indeed that sequestration of such drugs into lysosomes and endosomes is a mechanism of drug resistance and thus can be overcome by chloroquine.

What has not been previously understood was the mechanisms regulating this uptake and trapping. The authors here clearly show the redistribution of NHE6 from the endosomal compartment to the plasma membrane in mild hypoxia within a short period of time.

Comments

1 It would be interesting to know in a longer time frame what happens, does it continue to rise higher, does it plateau at 24 or 48 hours and in particular, how long does it stay up for? So is this effective in only transient hypoxia, chronic hypoxia and what effect does reoxygenation have? These are relatively straightforward experiments to carry out in tissue culture.

2 A key component previously studied has been P-glycoprotein which affects uptake but also redistribution into endosomes and I think it will be critical to have P-glycoprotein staining to go alongside this.

3 Even four hours of hypoxia is sufficient to induce many RNAs and it would be of interest to know whether RNAs for P-glycoprotein and NHE6 are also induced.

4 PKC has been reported by others to be activated by hypoxia but this is not clearly shown here that there is any phosphorylation or activation of downstream substrates, which would be important to demonstrate as the basic principle controlling the system described here.

5 Concomitantly with resistance to basic drugs there may be sensitisation to other drugs and so it would be of substantial interest to understand whether this caused the reciprocal effect on other drugs for which an acidic environment is helpful to uptake.

6 The study is focused on the intracellular acidic environment but it is potentially important that the extracellular environment might also be regulated by these changes. In this context, the paper last year by Gillies et al in Nature Communications on LAMP2 regulating the environment and having a major influence through the lysosomal compartment should be considered.

Does the endosomal compartment change concomitantly with the LAMP2 and is LAMP2 part of this system? I realise that there was chronic acidosis that induced LAMP2, but clearly this is a potential interaction of these two compartments under microenvironmental stress.

7 Two cell lines of different histological types are studied, which is commendable, but really a key issue is does this occur in normal tissues also. It would be useful to stain a small number of cancer

tissues with areas of hypoxia to show that it is not induced in the stroma but is induced in the cancer. Also, to use some non-malignant cell lines and although this is difficult for epithelium, MC10s are commonly used as well as HUVECs and fibroblasts lines.

8 Table 1 and table 2.

IC50 values on MTT assays are really not adequate to describe the drug sensitivity hence effects of the chemotherapy. The normal standard is clonogenic assays to show that cells that can go on for several generations are inhibited. I think it is essential these experiments are repeated with clonogenic assays.

Also, the IC50s in normoxia and hypoxia often have plateaus and I think the curve should be shown on the same axis for the 20% oxygen, 1% oxygen and the pH regulators. Then we can see where the median 50% line is drawn and whether the graphs have plateaued or not. Finally, MTT measures mitochondrial metabolism, which is surely not the best endpoint to use when you interfere with pH or oxygen concentrations?

9 Figure S2 needs more description. Clearly, under 1% oxygen, NHE6527588 shows a very different pattern to the control. Nevertheless, there is a strong membrane staining present and extensive endosomes present, which show different distribution to the controls.

The endosomes, if anything, seem more numerous and this would fit well with the idea that it stops the NHE6 going to the plasma membrane, but in fact it seems more numerous than in the control situation, thus the 1% oxygen and the peptide show more endosomes than the 21% oxygen with or without the peptide, so does this imply new synthesis? What happened to the mRNA? Or could this be protein stabilisation and the longer half life, which could be tested.

Overall, therefore, this paper shows a previously unknown pathway regulating endosomal pH, highly relevant to cancer therapy. They show could be selectively targeted, in this case they used a peptide and the appropriate controls.

However, the above experiments need to be done to complete the study and understand in better context the effect of this pathway, in tumour versus normal tissues, the timeframe and other pathways which are closely related and could be contributing to the experiments and results. Of most importance is to show that PKC is activated in hypoxia and that this activates a downstream signalling cascade.

.

Reviewer #2 (Remarks to the Author):

General comments: This is a comprehensive mechanistic study of the role of endosome sequestration as one of the factors leading to resistance to doxorubicin. The work is novel and interesting. In general the experiments are well designed and performed although on occasions the authors are overselling their effects, particularly in relation to inhibiting drug resistance by the blocking peptide.

Specific comments:

P2 lines 3-4: Reversal of drug transporters is a strategy that has not worked clinically (as they indicate elsewhere) – suggest delete from abstract.

P3 lines 1-6: Calling these drugs a mainstay of cancer treatment is inaccurate. While doxorubicin is still used, many other drugs with different mechanisms are also in use. Daunorubicin and mitoxantrone (which is an anthracenedione and not an anthracycline) are used very rarely to treat solid tumours. Same comment applies to p6, line 6. Please modify.

P6: Using IC50 endpoint in MTT assays is concealing important effects. It is known that hypoxia inhibits cell proliferation and the action of doxorubicin is cell cycle dependent. This alone will lead to resistance. The MTT assay (p20) requires 72-hour incubation and then assessment of viability (more correctly active metabolism) by dye uptake. Proliferation of cells in that 72 hour period will be less under hypoxia – so the control conditions vary as well as effects of Dox. Also bafilomycin is toxic but its inherent toxicity is not provided or discussed. These points need to be addressed – ideally the MTT assay should be supplemented by a colony-forming assay.

P9, line 6: While the volume of work is substantial the explanation for doing these experiments only on HT-1080 cells is weak – Δ pH is still substantial for the MBA-MD-231 cells and experiments were shown for these cells in Fig 4. If these experiments were done they should be reported, even if (or especially if) results were dissimilar. It is important to know if these effects are cell-line specific.

P13-14 and Fig 8: The idea of using an interfering peptide as a potential therapeutic agent is good but the effects are being greatly oversold. The CAM is not a very good model for solid tumours (especially using topical Dox, which has very poor penetration into tissue) and although the difference in size of the tumours with and without peptide is statistically significant, it is therapeutically unimportant; tumours grow exponentially and this size difference on a linear scale is trivial. The authors should be more critical of their data here.

P16, lines 6-7. In line with above comment – the authors cannot claim “to reverse drug resistance” – they have modified drug sensitivity very slightly.

Reviewer #3 (Remarks to the Author):

Lucien and co-workers present research in which drug resistance could be the result of sequestering of drug to acidified parts of the cancer cell. This effect is hypothesized to be pronounced in hypoxia and the mechanism proposed is mediated by sodium-hydrogen ion exchange channels which produce the pH gradient conducive for drug sequestering. This concept of pH-mediated drug resistance within cancer cells is not novel, but the mechanism the authors propose is novel and targetable. The authors show that they are able to control the impact of weakly basic chemotherapy drugs by inhibiting these sodium-hydrogen exchangers (NHE6/9) either with direct pharmacologic means or via its trafficking to the plasma membrane. Interfering with NHE6 trafficking to the plasma membrane was done by antagonizing its binding to RACK1 and PKC signalling.

The work is novel and impactful and provides greater molecular detail with regards to how drug resistance can occur via hypoxia-induced pH differentials within cancer cells. This work will be of high interest to those focused on drug resistance, in particular, the emphasis on hypoxia as a key feature in tumors that exhibit dysfunctional vasculature. The results support the key conclusions and while the second last paragraph borders on too much conjecture, the rest of the manuscript is valuable and does not over-interpret.

I have a number of questions/concerns:

1. Would it be possible to perform these experiments in conditioned media that is pH'd to 6, 6.5, 7? It may be important to consider the effect of extracellular interstitial fluid that is of a pH similar to that found during hypoxia. I would expect to see less uptake of the drug due to an accumulation at the extracellular space.
2. Is there a correlation with cells that have more lysosomal signal/organelles with increased resistance to weakly basic chemotherapy drugs? Flow cytometry could be used to confirm this.
3. Is over expression of NHE6 correlated with increased drug sensitivity?
4. Non-permeabilized cells for immunostaining of NHE6 should be performed as well if the antibody is specific for its extracellular epitope.
5. Does overexpression of RACK1 in hypoxia/normoxia result in pronounced effects in NHE6 plasma membrane localization?
6. If the IC50 occurs at higher concentrations in hypoxia, where does the excess drug compartmentalize to? What is the mechanism of drug sensitivity at those higher levels?
7. Please explain the lack of Pimo association with CAIX positive cancer cells in Figure 8C.

Secondary issues:

1. There is a small number of spelling mistakes or grammatical errors throughout.
2. Please provide rationale for using the Welch T-test in various tables, when a two-way ANOVA would have sufficed?

RESPONSE TO REVIEWERS

Reviewer #1 (Remarks to the Author):

Nature Communications 1108840 Commentary

Re. Hypoxia-induced mobilization of NHE6 to the plasma membrane by Lucien *et al.*

This paper shows that mild hypoxia induces mobilisation of the sodium-hydrogen exchanger NHE6 from endosomal compartment to the plasma membrane, dependent on PKC and markedly affecting endosomal pH. The translocation to the plasma membrane results in hyper acidification of early endosomes, which can then trap basic drugs such as doxorubicin, causing drug resistance.

These conventional anticancer drugs, topoisomerase II inhibitors, remain an important contribution to the treatment of many solid tumours, in spite of recent advances in non-chemotherapy approaches. It has long been known that drug resistance is mediated for basic drugs by an acidic microenvironment and indeed that sequestration of such drugs into lysosomes and endosomes is a mechanism of drug resistance and thus can be overcome by chloroquine.

What has not been previously understood was the mechanisms regulating this uptake and trapping. The authors here clearly show the redistribution of NHE6 from the endosomal compartment to the plasma membrane in mild hypoxia within a short period of time.

Comments

1-It would be interesting to know in a longer time frame what happens, does it continue to rise higher, does it plateau at 24 or 48 hours and in particular, how long does it stay up for? So is this effective in only transient hypoxia, chronic hypoxia and what effect does reoxygenation have? These are relatively straightforward experiments to carry out in tissue culture.

RESPONSE : We included new data showing the impact of re-oxygenation and longer exposure to hypoxia (24h) on plasma membrane NHE6 mobilization. Results presented in the **new Supplementary Figure 4 a,b** indicate that incubation of the cells for 24 h under hypoxic conditions results in a similar

increase in the distribution of NHE6 to the plasma membrane as compared to a shorter (4h) incubation time. In addition, re-exposure of the cells to ambient oxygen levels reduces plasma membrane NHE6 levels. These new data suggest that the mobilization of NHE6 to the plasma membrane is a sustain and reversible event.

2-A key component previously studied has been P-glycoprotein which affects uptake but also redistribution into endosomes and I think it will be critical to have P-glycoprotein staining to go alongside this. Even four hours of hypoxia is sufficient to induce many RNAs and it would be of interest to know whether RNAs for P-glycoprotein and NHE6 are also induced.

RESPONSE : Analysis of MDR1 mRNA expression showed undetectable levels in MDA-MB-231 cells exposed to normoxic or hypoxic conditions (**new supplementary Figure 7a**) The lack of P-glycoprotein in this cell line is consistent with previous studies (Bao L. et al., 2012, Am J of Pathology; Mutoh K. et al., 2006, Cancer Science).

The other cell line used in this study (HT-1080) express detectable levels of MDR1 mRNA but no increase was observed in cells incubated for 4h under low (1%) O₂ concentration (**new supplementary Figure 7b**). Because the 4h time point is the one used for Dox distribution studies, these results suggest that the increase in p-glycoprotein mRNA levels is not a mechanism by which Dox is redistributed within endosomes of cells exposed to *acute* hypoxia. Of note, a small but significant increases in *MDR1* mRNA expression was observed after 8h incubation in low O₂ which is consistent with a recent report showing that hypoxia can rapidly promote P-glycoprotein expression in laryngeal cancer cells (Li D. et al., 2016, Oncol Letters). Even though P-glycoprotein is unlikely to play a direct role in endosomal/lysosomal Dox distribution in acute hypoxia, we cannot exclude the potential contribution of P-glycoprotein in Dox distribution under sustain exposure to hypoxia as it is the case in solid tumors. These results are now discussed in the relevant part (**4th paragraph**) of the discussion.

For *SLC9A6* gene (or protein) expression, no significant differences were observed in HT-1080 or MDA-MB-231 cells within the time frame (4h and 8h incubation) and conditions (normoxia or hypoxia) used (results shown in the **updated Figure 4 d-g**). In contrast, the hypoxia-regulated gene *CA9* (positive control) was strongly upregulated.

Many studies have reported different subcellular localization of P-glycoprotein such as endosomes/lysosomes (Seebacher N. et al., 2016, Journal of Bio Chem) (Yamagishi T. et al., 2013, Journal of Bio Chem), plasma membrane (Katayama K. et al., 2016, BBA), nucleus (Szaflarski W. et al., 2013, Biomed Pharmacotherapy) and mitochondria (Shen Y. et al., 2012, Oncol Rep) indicating the protein may traffick between multiple subcellular compartments. It is therefore possible that P- glycoprotein is express on or traffick to the endosomal/lysosomal cell compartment under hypoxia, a feature that would suggest its participation in Dox sequestration. Confocal microscopy analysis of P- glycoprotein staining in HT-1080 cells indicates that the protein does not co-localize with EEA1+ endosomes or LAMP1+ lysosomes under normoxic or hypoxic conditions (**new Supplementary Figure 7b-d**; and **updated 3rd paragraph of the discussion**) The lack of regulation of P- glycoprotein (mRNA; trafficking) suggests that this transporter is not a key component of endosomal Dox sequestration under acute hypoxia.

3-PKC has been reported by others to be activated by hypoxia but this is not clearly shown here that there is any phosphorylation or activation of downstream substrates, which would be important to demonstrate as the basic principle controlling the system described here.

RESPONSE : To further show that PKC is activated in our system, we used a phospho-(Ser) PKC substrate antibody (Cell Signaling Technology #2261) in western blotting of cells lysates from cells incubated under hypoxic conditions or exposed to the PKC activator, PDBu. Data presented in the **new Figure 6 a,b** indicate that the phosphorylation of intracellular PKC substrates is increased by hypoxia with peak effect observed at 1h. Because RACK1 only binds to the activated form of PKC, our new results, in addition to the finding that PKC-RACK1 association is induced by hypoxia (Figure 6C), now clearly show that PKC is activated by hypoxia resulting in phosphorylation/activation of downstream substrates.

4-Concomitantly with resistance to basic drugs there may be sensitization to other drugs and so it would be of substantial interest to understand whether this caused the reciprocal effect on other drugs for which an acidic environment is helpful to uptake.

RESPONSE : Because weakly basic drugs are trapped in acidified endosomes of hypoxic cells, the uptake of weakly acid drugs such as Chlorambucil (pKa 5.78) and

Cyclophosphamide (pKa 6.0) should indeed be improved under similar pH conditions and we agree that comparison with these types of drug is of great interest.

However, we are not aware of reports that have tested the potential (or lack of) sequestration of weakly acid drugs within endosomes mostly because of technical limitations. In fact, weakly acid drugs do not have natural fluorescence (unlike Doxorubicin) to allow their tracking into cells so we are not in a position to do these experiments.

Another possibility could be to directly test the impact of endosomal pH modulation (hypoxia, RACK1-NHE6 blocking peptide and next-generation (cell permeant) peptide) on a panel of weakly basic neutral and weakly acid drugs in vitro and in tumors. However, the amount of work that we need to put on these experiments is not feasible within the Nature communications 3-month revision time-frame but will be feasible for a future and separate publication

5-The study is focused on the intracellular acidic environment but it is potentially important that the extracellular environment might also be regulated by these changes. In this context, the paper last year by Gillies et al in Nature Communications on LAMP2 regulating the environment and having a major influence through the lysosomal compartment should be considered. Does the endosomal compartment change concomitantly with the LAMP2 and is LAMP2 part of this system? I realise that there was chronic acidosis that induced LAMP2, but clearly this is a potential interaction of these two compartments under microenvironmental stress.

RESPONSE : Gillis *et al* in their Nature Communications paper (2015) indeed show that chronic acidosis induces the redistribution of LAMP2 to the plasma membrane in breast cancer. Even though they don't know the meaning or such redistribution or the exact mechanism involved, they proposed that plasma membrane LAMP2 redistribution could be part of an adaptive mechanism that protect the cells against acid-induced cell death and/or serves as a marker of acidosis. Even though this paper is of great interest, the potential mechanistic link between LAMP2 redistribution to the plasma membrane and our results showing Dox resistance under acute hypoxia is really not obvious especially considering that the exact role or mechanism of LAMP2 redistribution remains unknown. So, the design of experiments to address the role of LAMP2

redistribution in hypoxia-induced Dox resistance should await further knowledge on the mechanism and role of LAMP1 redistribution.

6- Two cell lines of different histological types are studied, which is commendable, but really a key issue is does this occur in normal tissues also. It would be useful to stain a small number of cancer tissues with areas of hypoxia to show that it is not induced in the stroma but is induced in the cancer. Also, to use some non-malignant cell lines and although this is difficult for epithelium, MC10s are commonly used as well as HUVECs and fibroblasts lines.

RESPONSE: Cell-surface biotinylation assays and immunofluorescence studies were performed using NHE6GFP-transfected cells so no NHE-6 specific Ab were involved in these assays. Very few Abs against NHE6 are available and the one that works in IHC or ICC targets the c-terminal intracellular portion of the exchanger so prevents visualization of plasma-membrane NHE6 in unpermeabilized tissues such as tumor sections.

7-Table 1 and table 2: IC50 values on MTT assays are really not adequate to describe the drug sensitivity hence effects of the chemotherapy. The normal standard is clonogenic assays to show that cells that can go on for several generations are inhibited. I think it is essential these experiments are repeated with clonogenic assays. Finally, MTT measures mitochondrial metabolism, which is surely not the best endpoint to use when you interfere with pH or oxygen concentrations?

RESPONSE : MTT reduction into formazan is performed by the mitochondrial enzyme succinate dehydrogenase and as mentioned by the reviewer, hypoxic conditions or reoxygenation may affect mitochondrial activity (Wang S. et al., 2011, Toxicol In Vitro). To circumvent this possibility, all incubations with MTT (including the hypoxic samples) were performed under normoxic conditions for 3h. For calculation of the IC_{50s}, we normalized both normoxic and hypoxic drug-treated samples with their untreated counterpart (see the **updated Materials and Methods** section on cell viability assays). Thus, potential biases were taken into consideration.

Nevertheless, we agree that clonogenic survival assay is the gold standard for the assessment of drug cytotoxicity. Unfortunately, and as previously shown (Franken NAP. et al., 2006, Nat Protocols), the cell lines used in this studies (HT-1080 and MDA-MB-231) already have very low plating efficiency so our numerous

attempts to use this assays for measuring Dox toxicity did not yield reliable results since to few colonies were remaining after treatments. Data obtained from the MTT assays were therefore confirmed by direct cell counting of viable cells using the trypan blue exclusion method (**new Supplementary Fig.1 g-j**).

8- Also, the IC50s in normoxia and hypoxia often have plateaus and I think the curve should be shown on the same axis for the 20% oxygen, 1% oxygen and the pH regulators. Then we can see where the median 50% line is drawn and whether the graphs have plateaued or not.

RESPONSE : As suggested by the reviewer, we added a new figure (**Supplementary Figure 1a-f**) that shows the viability curves for the MTT assays. This new information clearly shows that all results were performed using experimental conditions that are suitable for accurate calculation of IC50s.

9-Figure S2 needs more description. Clearly, under 1% oxygen, NHE6⁵²⁷⁵⁸⁸ shows a very different pattern to the control. Nevertheless, there is a strong membrane staining present and extensive endosomes present, which show different distribution to the controls.

The endosomes, if anything, seem more numerous and this would fit well with the idea that it stops the NHE6 going to the plasma membrane, but in fact it seems more numerous than in the control situation, thus the 1% oxygen and the peptide show more endosomes than the 21% oxygen with or without the peptide, so does this imply new synthesis? What happened to the mRNA? Or could this be protein stabilization and the longer half-life, which could be tested.

RESPONSE : We went back to our data (pictures) from the immunofluorescence studies and even though the number of EEA1 positive endosomes found in cells varies from cell to cell, the average size or number of endosomes under the experimental conditions used (acute hypoxia, presence or absence of the NHE6 peptide) does not vary (see **new Figure 1 e,f**). We therefore changed the representative pictures accordingly (**updated Supplementary figure 5**).

Overall, therefore, this paper shows a previously unknown pathway regulating endosomal pH, highly relevant to cancer therapy. They show could be selectively targeted, in this case they used a peptide and the appropriate controls. However, the above experiments need to be done to complete the study and understand in better context the effect of this pathway, in tumour versus normal tissues, the timeframe and other pathways which are closely related and could be

contributing to the experiments and results. Of most importance is to show that PKC is activated in hypoxia and that this activates a downstream signalling cascade.

Reviewer #2 (Remarks to the Author):

General comments: This is a comprehensive mechanistic study of the role of endosome sequestration as one of the factors leading to resistance to doxorubicin. The work is novel and interesting. In general the experiments are well designed and performed although on occasions the authors are overselling their effects, particularly in relation to inhibiting drug resistance by the blocking peptide.

Specific comments:

-P2 lines 3-4: Reversal of drug transporters is a strategy hat has not worked clinically (as they indicate elsewhere) – suggest delete from abstract.

RESPONSE : The abstract has been modified accordingly to reviewer’s comment.

-P3 lines 1-6: Calling these drugs a mainstay of cancer treatment is inaccurate. While doxorubicin is still used, many other drugs with different mechanisms are also in use. Daunorubicin and mitoxantrone (which is an anthracenedione and not an anthracycline) are used very rarely to treat solid tumours. Same comment applies to p6, line 6. Please modify.

RESPONSE : We modified this statement for the following : « Commonly used therapy for the management of many cancers includes chemotherapy regimens based anthraquinone derived drugs (anthracycline: doxorubicin, daunorubicin) (anthracenedione: mitoxantrone)”

-P6: Using IC50 endpoint in MTT assays is concealing important effects. It is known that hypoxia inhibits cell proliferation and the action of doxorubicin is cell cycle dependent. This alone will lead to resistance. The MTT assay (p20) requires 72-hour incubation and then assessment of viability (more correctly active metabolism) by dye uptake. Proliferation of cells in that 72 hour period will be less under hypoxia – so the control conditions vary as well as effects of Dox.

RESPONSE: The reviewer is right, we observed less proliferation of cells under hypoxic conditions but this bias has been taken into account. For calculation of

the IC_{50s}, we normalized both normoxic and hypoxic drug-treated samples with their untreated counterpart (see the **updated Materials and Methods** section on cell viability assays).

- Also bafilomycin is toxic but its inherent toxicity is not provided or discussed. These points need to be addressed –

RESPONSE: We added a statement in the discussion section (**updated 3rd paragraph**) related to the toxicity of bafilomycinA ,which prevents its clinical application.

- ideally the MTT assay should be supplemented by a colony-forming assay.

RESPONSE: As answered to reviewer 1, , we agree that clonogenic survival assay is the gold standard for the assessment of drug cytotoxicity. Unfortunately, and as previously shown (Franken NAP. et al., 2006, Nat Protocols), the cell lines used in this studies (HT-1080 and MDA-MB-231) already have very low plating efficiency so our numerous attempts to use this assays for measuring Dox toxicity did not yield reliable results since to few colonies were remaining after treatments. Data obtained from the MTT assays were therefore confirmed by direct cell counting of viable cells using the trypan blue exclusion method (**new Supplementary Fig.1 g-j**).

-P9, line 6: While the volume of work is substantial the explanation for doing these experiments only on HT-1080 cells is weak – Δ pH is still substantial for the MBA-MD-231 cells and experiments were shown for these cells in Fig 4. If these experiments were done they should be reported, even if (or especially if) results were dissimilar. It is important to know if these effects are cell-line specific.

RESPONSE : We now present new results showing that:

- 1- Silencing of NHE6 in HT-1080 or MDA-MB-231 cells results in hyperacidification of the intravesicular compartment to a level similar to that induced by hypoxia, whereas depletion of the NHE9 isoform had no significant impact (**Fig. 3a-e and new Supplementary Fig. 2 a,c**).
- 2- Overexpression of NHE6 in MDA-MB-231 increases the pH of intracellular vesicles and blocks the acidification induced by hypoxia (**new Supplementary Fig.2e**).
- 3- PKC is activated by hypoxia in both HT-1080 and MDA-MB-231 cells (**New Figure 6a,b**)

-P13-14 and Fig 8: The idea of using an interfering peptide as a potential therapeutic agent is good but the effects are being greatly oversold. The CAM is not a very good model for solid tumours (especially using topical Dox, which has very poor penetration into tissue) and although the difference in size of the tumours with and without peptide is statistically significant, it is therapeutically unimportant; tumours grow exponentially and this size difference on a linear scale is trivial. The authors should be more critical of their data here.

RESPONSE : Thanks for this comment. We agree that the changes in Dox sensitivity of tumors derived from NHE6⁵²⁷⁻⁵⁸⁸ peptide overexpressing cells are small, even though significant, and it was certainly not our intention to oversell these results. We therefore modified the relevant results section accordingly: -“Treatment of NHE6⁵²⁷⁻⁵⁸⁸-overexpressing HT-1080 or MDA-MB-231 tumor xenografts with these suboptimal concentrations of Dox lead to a *small but* significant decrease in tumor volume compared to xenografts overexpressing a control peptide (Fig. 8f-h).”

We also changed the title of this section:“NHE6-RACK1 blockade *partially* restores Dox sensitivity”

- P16, lines 6-7. In line with above comment – the authors cannot claim “to reverse drug resistance” – they have modified drug sensitivity very slightly.

RESPONSE : We have changed the statement « our ability to reversed drug resistance » to « our ability to *partially* reverse drug resistance », a statement that is indeed more in line with the results.

Reviewer #3 (Remarks to the Author):

Lucien and co-workers present research in which drug resistance could be the result of sequestering of drug to acidified parts of the cancer cell. This effect is hypothesized to be pronounced in hypoxia and the mechanism proposed is mediated by sodium-hydrogen ion exchange channels which produce the pH gradient conducive for drug sequestering. This concept of pH-mediated drug resistance within cancer cells is not novel, but the mechanism the authors propose is novel and targetable. The authors show that they are able to control the impact of weakly basic chemotherapy drugs by inhibiting these sodium-hydrogen exchangers (NHE6/9) either with direct pharmacologic means or via its trafficking to the plasma membrane. Interfering with NHE6 trafficking to the plasma membrane was done by antagonizing its binding to RACK1 and PKC

signalling.

The work is novel and impactful and provides greater molecular detail with regards to how drug resistance can occur via hypoxia-induced pH differentials within cancer cells. This work will be of high interest to those focused on drug resistance, in particular, the emphasis on hypoxia as a key feature in tumors that exhibit dysfunctional vasculature. The results support the key conclusions and while the second last paragraph borders on too much conjecture, the rest of the manuscript is valuable and does not over-interpret.

RESPONSE: The discussion related to the potential role of NHERF1 in plasma-membrane localization of NHE6 (last second paragraph) is indeed not supported by results presented in this manuscript so this section has been deleted from the revised version.

I have a number of questions/concerns:

1. Would it be possible to perform these experiments in conditioned media that is pH'd to 6, 6.5, 7? It may be important to consider the effect of extracellular interstitial fluid that is of a pH similar to that found during hypoxia. I would expect to see less uptake of the drug due to an accumulation at the extracellular space.

RESPONSE: Extracellular acidification is indeed an important hallmark of chemoresistance to weakly basic drugs and pH-based therapy has demonstrated promising results in vitro and in murine models (Wojtkowiak JW. et al., 2012, Mol Pharm)(Gerweck LE. et al., 2006, MCT). We therefore followed the reviewer's suggestion and incubated HT-1080 cells in media of different pHs. The results show (**new supplementary Figure 6**) that extracellular acidification reduces doxorubicin uptake as observed by the decrease in fluorescence intensity of the nucleus. Our study suggest that extracellular acidification is a barrier to weak-base drug uptake and once inside the cells, hypoxia-induced pH partitioning of drugs within endosomes represents an additional resistance mechanism. This is now discussed in the **updated 2nd paragraph of the discussion** section.

2. Is there a correlation with cells that have more lysosomal signal/organelles with increased resistance to weakly basic chemotherapy drugs? Flow cytometry could be used to confirm this.

RESPONSE: it is indeed interesting to consider endosome and lysosome biogenesis as a potential mechanism of chemoresistance through drug sequestration and secretion. Using confocal microscopy we counted the number

of EEA1 positive and LAMP2 positive vesicles per cells. The results did not show any differences in the number of endosomes and lysosomes in cells cultured under hypoxic conditions compared to normoxia (see **updated Figure 1 e,f**) suggesting that the biogenesis of vesicles is not a mechanism explaining the increased sequestration/resistance to Dox in our study.

3. Is over expression of NHE6 correlated with increased drug sensitivity?

RESPONSE : Overexpression of NHE6 in MDA-MB-231 cells increases the pH of intracellular vesicles and blocks the acidification induced by hypoxia. (**new Supplementary Fig.2e**).The results suggest that overexpression of NHE6 should decrease weakly basic drug trapping and increase sensitivity.

4. Non-permeabilized cells for immunostaining of NHE6 should be performed as well if the antibody is specific for its extracellular epitope.

RESPONSE : Cell-surface biotinylation assays and immunofluorescence studies were performed using NHE6GFP-transfected cells so no NHE-6 specific Ab were involved in these assays. Very few Abs against NHE6 are available and the one that works in IHC or ICC targets the c-terminal intracellular portion of the exchanger so prevents visualization of plasma-membrane NHE6 in unpermeabilized cells.

5. Does overexpression of RACK1 in hypoxia/normoxia result in pronounced effects in NHE6 plasma membrane localization?

RESPONSE : In the various tissues and cell lines tested, RACK1 was found to be a very abundant scaffold protein so we did not consider doing overexpression studies as we believe that such experiments are unlikely to yield conclusive results.

6. If the IC50 occurs at higher concentrations in hypoxia, where does the excess drug compartmentalize to? What is the mechanism of drug sensitivity at those higher levels?

RESPONSE : According to the pH-dependent drug partitioning theory, weakly basic drugs will be partitioned into acidified compartments of hypoxic cells, mostly endosomal vesicles, as shown in this paper. Weakly basic drugs will also be expelled to the cell supernatant presumably through exocytosis. This latter

possibility is difficult to assess when the extracellular concentration of drug is already high.

Mechanisms of drug resistance under hypoxic conditions include a decrease of drug uptake inside cells and RACK/PKC-dependent sequestration of NHE6 exchanger as shown in this paper. It is the interference with such segregation mechanism (through PKC inhibition or RACK-1/NHE6 inhibition peptide) that confers weak base drug sensitization in hypoxic cells.

7. Please explain the lack of Pimo association with CAIX positive cancer cells in Figure 8C.

RESPONSE : The fact that CA9 displayed greater areas of staining than pimonidazole is consistent with earlier reports showing differences between the pO₂ dependency of pimonidazole binding and CA9 protein expression. This is now explained in the revised version of the manuscript (**result section; page 13, lines 14-17**)

Secondary issues:

1. There is a small number of spelling mistakes or grammatical errors throughout.

RESPONSE: typos and grammatical errors have been corrected throughout the manuscript.

2. Please provide rationale for using the Welch T-test in various tables, when a two-way ANOVA would have sufficed?

RESPONSE : Statistical data analysis has been performed on IC₅₀ values (not viability curves) obtained with several independent experiments. We used the unpaired t Test with Welch correction because some of our compared samples have normal distributions but unequal variances.

REVIEWERS' COMMENTS:

Reviewer #1 (Remarks to the Author):

The questions are adequately answered, although clonogenic assays could not be done, over endpoints were measured.

Reviewer #2 (Remarks to the Author):

I have no further comments and am satisfied that the authors have responded adequately to concerns raised in the previous review.

Reviewer #3 (Remarks to the Author):

The authors have addressed all of my previous concerns and questions.